# `LoRDO`: Distributed Low-Rank Optimization with Infrequent Communication

**Andrej Jovanović** [1 2]  **Alex Iacob** [1 2]  **Mher Safaryan** [3]  **Ionut-Vlad Modoranu** [4]  **Lorenzo Sani** [1 2]  **William F. Shen** [1]
**Xinchi Qiu** [1]  **Dan Alistarh** [4 5]  **Nicholas D. Lane** [1 2]

## Abstract

Distributed training of foundation models via `DDP` is limited by interconnect bandwidth. While infrequent communication strategies reduce synchronization frequency, they remain bottlenecked by the memory and communication requirements of optimizer states. Low-rank optimizers can alleviate these constraints; however, in the local-update regime, workers lack access to the full-batch gradients required to compute low-rank projections, which degrades performance. We propose `LoRDO`, a principled framework unifying low-rank optimization with infrequent synchronization. We first demonstrate that, while global projections based on pseudo-gradients are theoretically superior, they permanently restrict the optimization trajectory to a low-rank subspace. To restore subspace exploration, we introduce a full-rank quasi-hyperbolic update. `LoRDO` achieves near-parity with low-rank `DDP` in language modeling and downstream tasks at model scales of 125M–720M, while reducing communication by $\approx 10\times$. Finally, we show that `LoRDO` improves performance even more in very low-memory settings with small rank/batch size.

## 1. Introduction

Distributed optimization methods, such as DiLoCo ([Douillard et al., 2023](#)) and DES-LOC ([Iacob et al., 2026b](#)), have emerged as a solution to the substantial communication overheads inherent to Distributed Data Parallel (`DDP`) training. By leveraging local updates, these approaches reduce the bandwidth requirements for training large models.

However, the deployment of such methods faces *two critical constraints*. First, for large-scale training ([Brown et al.,](#) 2020; [Touvron et al., 2023](#); [Dubey et al., 2024](#); [DeepSeek-AI et al., 2025](#)), the local optimization procedure on each worker incurs a significant memory overhead when storing optimizer momenta ([Kingma & Ba, 2015](#); [Pagliardini et al., 2025](#); [Jordan et al., 2024](#)), limiting the maximum trainable model size. Second, communicating optimizer states, required for convergence guarantees ([Cheng & Glasgow, 2025](#)), undermines the communication-efficiency gains of infrequent synchronization ([Douillard et al., 2023](#)).

Low-rank adaptive optimizers, such as `GaLore` ([Zhao et al., 2024](#)) and `LDAdam` ([Robert et al., 2025](#)), offer a path to alleviate the memory and communication bottlenecks of large-scale training. However, generalizing these methods to the infrequent-synchronization regime, while preserving guarantees, presents an optimization challenge. We show that computing low-rank projections locally on individual workers is detrimental; the reduced effective batch size per data shard introduces significant projection noise, resulting in a suboptimal projection subspace. While a global projection strategy, derived from the aggregated pseudo-gradient, can mitigate this noise and recover a stable basis analogous to `DDP`, we show that it introduces a critical failure mode: *it permanently constrains the optimization to a fixed rank-$r$ subspace*. This rank restriction severely limits exploration, reducing final performance during training. To resolve this, we investigate the following design question:

> *How can we design high-performance low-rank optimizers for communication-efficient training?*

To answer this, we propose `LoRDO`, a framework that adapts low-rank optimizers for distributed training with infrequent communication. Specifically, we demonstrate that injecting a full-rank quasi-hyperbolic momentum signal into each worker's update prevents stagnation of the global projection. This modification allows `LoRDO` to have near-parity with `DDP` and full-rank baselines while retaining the efficiency benefits of low-rank structures. Empirically, `LoRDO` reduces the communication overhead of low-rank `DDP` by $\approx 10\times$ at the 125M and 720M model scales. Despite these substantial reductions, `LoRDO` maintains near-parity with this baseline, exhibiting a negligible perplexity gap of less than $1\%$ and matched downstream task accuracy. The contributions of our work are as follows:

---

[1]University of Cambridge [2]Flower Labs [3]Lancaster University [4]Institute of Science and Technology Austria [5]Red Hat AI. Correspondence to: Andrej Jovanović <aj693@cam.ac.uk>.

*Proceedings of the $43^{rd}$ International Conference on Machine Learning*, Seoul, South Korea. PMLR 306, 2026. Copyright 2026 by the author(s).

**Contributions :**

1. **Analysis of projection failure modes.** We demonstrate that local projections harm performance due to high variance arising from small worker batch sizes, while global projections induce subspace stagnation.

2. **Restoring full subspace exploration.** We propose LoRDO, a low-rank optimizer which injects a full-rank gradient signal into the local update while maintaining a global projection derived from the aggregated pseudo-gradient. This prevents stagnation without increasing communication/memory overheads.

3. **DDP parity and efficiency.** We show that LoRDO achieves near-parity with synchronous low-rank DDP (perplexity gap $< 1\%$) while reducing optimizer memory and communication by up to $8 \times - 12 \times$. Under heavy memory constraints, which necessitate low ranks, LoRDO surpasses DDP by $3.36 - 4.7\%$ in perplexity, while also demonstrating superior resilience to small-batch regimes compared to local projection methods.

## 2. Low-Rank Adaptive Optimization

As a motivating example, we describe low-rank optimizers using a single linear layer $W \in \mathbb{R}^{p \times q}$, a core component of Transformer models we use in Section 4. Consider training a model $x$ with $M$ workers. Using Adam, each worker $m$ computes the following at time step $t$:

$$G_t^m \leftarrow \nabla F(x_t^m; \xi_t^m)$$
$$u_t^m \leftarrow \beta_1 u_{t-1}^m + (1 - \beta_1) G_t^m$$
$$v_t^m \leftarrow \beta_2 v_{t-1}^m + (1 - \beta_2)(G_t^m \odot G_t^m)$$
$$x_{t+1}^m \leftarrow x_t^m - \eta \frac{\hat{u}_t^m}{\sqrt{\hat{v}_t^m} + \epsilon}$$

Each worker stores two optimizer states of size $O(pq)$, equal to the local gradient size. For large-scale models, this creates a significant memory bottleneck. Low-rank adaptive optimizers (Zhao et al., 2024; Robert et al., 2025) alleviate this by maintaining momenta in a projected low-rank form while allowing full solution exploration. Using a projection matrix $Q_t^m : \mathbb{R}^{p \times r}$, the update becomes:

$$g_t^m \leftarrow (Q_t^m)^\top \nabla F(x_t^m; \xi_t^m)$$
$$u_t^m \leftarrow \beta_1 u_{t-1}^m + (1 - \beta_1) g_t^m$$
$$v_t^m \leftarrow \beta_2 v_{t-1}^m + (1 - \beta_2)(g_t^m \odot g_t^m)$$
$$x_{t+1}^m \leftarrow x_t^m - \eta Q_t^m \left( \frac{\hat{u}_t^m}{\sqrt{\hat{v}_t^m} + \epsilon} \right).$$

This regime reduces optimizer state memory overhead from $O(2pq)$ to $O(2r(p + q))$, where typically $r \ll p, q$.

To compute the projection matrix $Q$, Zhao et al. (2024) employ periodic SVD updates (Golub & Reinsch, 1970),

while Robert et al. (2025) use PowerSGD (Vogels et al., 2019) to estimate singular vectors at every step. An SVD projection in the DDP regime at step $t$ for worker $m$ is:

$$U, S, V \leftarrow SVD(G_t^m)$$
$$Q_t^m \leftarrow U[:, :r].$$

## 3. Designing LoRDO

In this section, we describe the key design decisions behind LoRDO, whose pseudocode is given in Algorithm 1. We first show that the global projection approach, while theoretically superior, can restrict learning to a stagnant subspace (see Section 5.1). LoRDO adds a full-rank quasi-hyperbolic momentum term that restores full subspace exploration while realizing the initial theoretical benefits, bringing empirical improvements (Section 5.2). Additionally, we outline that aligned momenta (Robert et al., 2025) and error feedback (Seide et al., 2014) are essential for optimal performance, as ablated in Figures 7 and 8. Finally, we provide a discussion on the memory and communication benefits achieved by LoRDO, which is elaborated further in Sections A.5 and A.6.

**Notation.** We consider standard distributed training settings with $M$ workers, where each worker performs $K$ local updates prior to synchronization. Training is conducted by minimizing a global objective function $f(x) := \frac{1}{M} \sum_{m=1}^{M} f_m(x)$ over the model parameters $x$, where each $f_m(x)$ is the local objective $\mathbb{E}_{\xi \sim \mathcal{D}_m}[F_m(x; \xi)]$, where the $F_m$ is the local loss for a data sample $\xi$ drawn from data distribution $\mathcal{D}_m$. Full derivations are provided in Section A.

### 3.1. LoRDO Projection Matrices

In DDP, all workers use a shared projection matrix $Q_t^m$ as gradients are synchronized across all workers prior to the optimizer step. However, in distributed optimization schemes such as those introduced in Douillard et al. (2023); Iacob et al. (2026b), parameter and optimizer state synchronization occurs only after $K$ steps of local training. Adapting low-rank optimizers to the local-update regime is non-trivial as workers lack access to the full-batch gradients required to compute projection matrices $Q_m^t$ as in DDP.

The naïve integration of low-rank optimizers into such frameworks is to allow each worker $m$ to determine its own projection matrix $Q_t^m$ *locally* based on its stochastic gradient $G_t^m$. However, we now discuss two issues related to this approach, which we resolve by the introduction of a global projection based on the aggregated pseudo-gradient.

**Lack of Worker Unification.** Since each worker determines its own projection matrix $Q_t^m$, workers are not guaranteed to optimize within the same basis, as each individual projection matrix could isolate an indepen-

---

**Algorithm 1** `LoRDO-Global`- Bias Correction Omitted for Ease of Notation

---

**Require: Model tensors, hyper-parameters**

1:    $T, M \in \mathbb{N}_+$ — total optimization steps and number of workers
2:    $\beta_1, \beta_2 \in [0, 1), \omega \in [0, 1]$ — decay rates for each momentum state and QHM convex combination coefficients
3:    $\rho \in \mathbb{R}_+, \{\eta_t\}_{t=0}^{T-1}$ — clipping radius, learning-rate schedule
4:    $K_x, K_u, K_v \in \mathbb{N}_+$ — communication periods for parameters and states
5:    `OuterOpt` $: (\mathbb{R}^d, \mathbb{R}^d) \to \mathbb{R}^d$ — update params using an outer optimizer, averaging by default
6:    `ComputeProjection` $: (\mathbb{R}^{d \times d}, \mathbb{R}) \to \mathbb{R}^{d \times r}$ — Compute projection routine (by default SVD)
7:    $x_0^m = x_0 \in \mathbb{R}^d, u_{-1}^m = \mathbf{0}_r, v_{-1}^m = \mathbf{0}_r$ — initial params, first and second momentum
8:    $Q_0 : \mathbb{R}^{d \times r}, E_{-1}^m = \mathbf{0}_{d \times d}, \forall m \in M$ - Random initial projection matrix and zeroed-out error buffer for each client

**Ensure:** $x_T, u_{T-1}, v_{T-1}$

9:    **for** $t = 0, \ldots, T - 1$ **do**
10:      **for all** workers $m = 0, \ldots, M - 1$ **in parallel do**
11:          $\hat{G}_t^m \leftarrow \mathbf{clip}(\nabla F(x_t^m; \xi_t^m), \rho)$             Clipped stochastic gradient in full-rank
12:          $\hat{g}_t^m \leftarrow Q_t^\top (\hat{G}_t^m + E_{t-1}^m)$             Low-rank gradient signal with error-feedback
13:          $E_t^m \leftarrow \hat{G}_t^m + E_{t-1}^m - Q_t \hat{g}_t^m$             Compute error feedback
14:          $u_t^m \leftarrow \beta_1 u_{t-1}^m + (1 - \beta_1)\hat{g}_t^m$
15:          $v_t^m \leftarrow \beta_2 v_{t-1}^m + (1 - \beta_2)(\hat{g}_t^m)^2$

16:         
$$\bar{x}_t^m \leftarrow x_t^m - \eta_t \begin{cases} Q_t \left[ \dfrac{u_t^m}{\sqrt{v_t^m} + \epsilon} \right] & \textcolor{red}{\text{No QHM}} \\[2ex] Q_t \left[ \dfrac{\omega u_t^m + (1-\omega)\hat{g}_t^m}{\sqrt{v_t^m} + \epsilon} \right] & \textcolor{red}{\text{Low-Rank QHM}} \\[2ex] (1-\omega)\dfrac{\hat{G}_t^m}{\mu(\sqrt{v_t^m} + \epsilon)} + \omega Q_t \left[ \dfrac{u_t^m}{\sqrt{v_t^m} + \epsilon} \right] & \textcolor{red}{\text{Full-Rank QHM}} \end{cases}$$

17:          $\bar{u}_t^m \leftarrow$ **if** $((t+1) \bmod K_u = 0)$ **then** $\mathbb{E}_m[u_t^m]$ **else** $u_t^m$      Sync $u$ every $K_j$
18:          $\bar{v}_t^m \leftarrow$ **if** $((t+1) \bmod K_v = 0)$ **then** $\mathbb{E}_m[v_t^m]$ **else** $v_t^m$      Sync $v$ every $K_v$
19:          **if** $((t+1) \bmod K_x = 0)$ **then**             Sync $x$ every $K_x$
20:             $\Delta_t^m \leftarrow \bar{x}_t^m - x_{t-K_x}^m; \Delta_t \leftarrow \mathbb{E}_m[\Delta_t^m]$      Compute per-worker and aggregated pseudo-gradient
21:             $x_{t+1}^m \leftarrow$ `OuterOpt`$(\Delta_t, x_{t-K_x}^m)$      New model update on previous model copy with aggregated pseudo-gradients.
22:             $Q_{t+1} \leftarrow$ `ComputeProjection`$(\Delta_t)$      Compute a new global projection matrix
23:             $u_t^m \leftarrow Q_{t+1}^\top Q_t \bar{u}_t^m$      Rotate the first moment locally
24:             $v_t^m \leftarrow (1 - \beta_2^t) \left| (Q_{t+1}^\top Q_t)^2 (\hat{\bar{v}}_t^m - (\hat{\bar{u}}_t^m)^2) + (Q_{t+1}^\top Q_t \hat{\bar{u}}_t^m)^2 \right|$      Rotate the second moment locally
25:          **else**
26:             $Q_{t+1} \leftarrow Q_t$      Maintain previous projection
27:             $x_{t+1}^m \leftarrow \bar{x}_t^m$      Maintain local model
28:             $u_t^m \leftarrow \bar{u}_t^m; \quad v_t^m \leftarrow \bar{v}_t^m$

---

dent subspace. This causes interference upon aggregating the pseudo-gradients: $\sum_{m=1}^M \sum_{\tau=t-k}^t \eta_t^m Q_m^t \alpha_t^m \neq \bar{Q}_t \sum_{m=1}^M \sum_{\tau=t-K}^t \eta_t^m \alpha_t^m$ where $\bar{Q}_t$ is the projection matrix that would have been obtained if using a gradient averaged across all workers as in `DDP`. While in the `IID` case, this may slightly lower final performance, for `Non-IID` data distributions, it may cause complete divergence.

**Lower-Quality Projections.** Assume that the stochastic gradient $\hat{G}$ is a perturbed version of the true gradient $G$ such that $\hat{G} = G + E$, where $E$ represents the additive noise incurred through the stochastic sample (Bottou et al., 2018). Furthermore, we assume that the noise scales proportionally to $\frac{\kappa}{\sqrt{B}}$, where $B$ is the batch size and $\kappa$ is the variance of

the individual samples (McCandlish et al., 2018; Bottou et al., 2018). Additionally, as showed by Xie et al. (2023), we assume that the singular values of $G$ and $\hat{G}$ follow a power-law where $\sigma_r = Cr^{-\alpha}$ where $\alpha > 0$. Using the Davis-Kahan $\sin \Theta$ theorem (Stewart & Sun, 1993; Davis & Kahan, 1970), we derive the instability of the projection matrix:

$$\Delta(\hat{Q}) \approx \frac{\kappa/\sqrt{B}}{\alpha C r^{-(\alpha+1)}} = \frac{\kappa}{\alpha C \sqrt{B}} \cdot r^{\alpha+1} \tag{1}$$

Examining $\Delta(\hat{Q})$, we see that the instability of the projection matrix is $O(B^{-0.5})$. When comparing `DDP` and local gradients: `DDP`'s gradient has an effective batch size of $MB$; the per-worker gradient with local batch size $B$ is

aggregated across $M$ workers. When using local gradients, however, the effective batch size is the local batch size. Determining projections locally yields a significantly noisier approximation than `DDP`. We also observe a dependence on the choice of rank $r$. In cases where $r \ll p, q$, there is less instability in the projection as it captures the most important dimensions of the signal, ignoring the noisy tail. As $r \to m, n$, the projection becomes more unstable due to noise affecting the basis estimation, providing a mathematical insight into the regularization induced by compression observed by Robert et al. (2025). Our results in Section 5.3 show that smaller batch sizes disproportionately impact local methods due to this noise sensitivity.

**Global Projections as a Solution.** Instead, we propose to use *global* projections that are shared across all workers at the synchronization boundary; this guarantees that all workers optimize within the same subspace. Specifically, $Q_t$ is computed from the aggregated pseudo-gradient $\Delta_t$, which represents the total change in model parameters following $K$ local optimization steps. Furthermore, as in `DDP`, the pseudo-gradient has an effective batch size of $MB$, because it is aggregated across workers, yielding a more stable signal with reduced variance. We further posit that the pseudo-gradient is more informative as it contains curvature information baked into the pseudo-gradient signal, which is not available by purely observing the local gradient. We present a more detailed discussion of this in Section D.8.

### 3.2. Enabling Full Subspace Exploration

Although the global projection in Section 3.1 is theoretically superior, as it unifies worker optimization directions and leverages a higher-quality projection basis, it is guaranteed to restrict learning to a stagnant subspace. We propose that adding a full-rank quasi-hyperbolic momentum term alleviates this stagnation by injecting a full-rank signal into the pseudo-gradient, allowing full subspace exploration.

**Stagnant Learning.** Computing the aggregated pseudo-gradient after a $K$ window of local training with low-rank optimization (derivations in Section A.3):

$$\underbrace{\Delta_t \leftarrow \frac{1}{|M|} \sum_{m=1}^{M} \overbrace{\sum_{\tau=t-K}^{t} \eta_\tau^m Q_\tau^m \alpha_\tau^m}^{\text{Pseudo-gradient}}}_{\text{Local}} \quad \underbrace{\Delta_t \leftarrow \frac{1}{|M|} Q_t \sum_{m=1}^{M} \overbrace{\sum_{\tau=t-K}^{t} \eta_\tau^m \alpha_\tau^m}^{\text{Pseudo-gradient}}}_{\text{Global}}$$

In this form, we observe `LoRDO-Global` effectively truncates the aggregated pseudo-gradient $\Delta_t$ to an $r$-rank subspace defined by the global projection matrix. Any optimization step on $\Delta_t$ is guaranteed to use a signal that is at most of rank $r$, reducing the possible optimization directions. Moreover, every new projection computed from this pseudo-gradient signal returns the same rank $r$ subspace.

`LoRDO-Local` does not suffer from the same pathology: the summation of mutually orthogonal projection matrices recovers the full representation (Horn & Johnson, 1985). As such, both `DDP` and the *local* variant can fully explore the solution space.

**Full-Rank Quasi-Hyperbolic Momenta.** We posit that applying quasi-hyperbolic momentum terms to `LoRDO-Global` will bypass this aforementioned pathology by injecting a full-rank signal into the pseudo-gradient update, in addition to improving performance (Iacob et al., 2026a). In low-rank optimizers, quasi-hyperbolic momentum terms can be applied in one of two forms, where $\mu(z) = \frac{1}{r} \sum_{i=1}^{r} z_i$:

$$Q_t^m \left[ \frac{(\omega u_t^m + (1-\omega)\hat{g}_t^m)}{\sqrt{v_t^m} + \epsilon} \right] \quad \text{Low-Rank QHM}$$

$$(1-\omega)\frac{\hat{G}_t^m}{\mu(\sqrt{\hat{v}_t^m} + \epsilon)} + \omega Q_t^m \left[ \frac{u_t^m}{\sqrt{v_t^m} + \epsilon} \right] \quad \text{Full-Rank QHM}$$

In its low-rank form, the quasi-hyperbolic moment is constrained to the same low-rank basis of the global projection matrix. However, by applying the gradient signal following the up-projection, scaled based on the second momentum, a full-rank signal is injected into the pseudo-gradient. Mathematically, this prevents the aggregated pseudo-gradient from remaining trapped within a fixed rank-$r$ subspace; instead, it recovers the full-rank representation given a sufficient number of workers ($M \times r \geq \min(p, q)$, assuming $r$ is the rank of the pseudogradient). Consequently, this enables the optimization procedure to temporally aggregate these rank-$(M \times r)$ subspaces, *enabling eventual complete subspace exploration* in the spirit of GaLore (Zhao et al., 2024).

Furthermore, we note that the addition of the quasi-hyperbolic momentum term serves a dual purpose in `LoRDO`. As mentioned above, its primary benefit is that it enables full-subspace exploration. A secondary advantage is that it allows the momentum half-lives to safely match their synchronization intervals, consistent with Iacob et al. (2026a).

**Additional Considerations.** Following Robert et al. (2025), we always rotate momenta following the computation of a new projection matrix to ensure that momentum updates are always accumulated on the same subspace. Additionally, we use error-feedback locally, similar to Robert et al. (2025); Seide et al. (2014), to improve the performance of the local optimization procedure. We provide an ablation for both of these aspects in Figures 7 and 8, respectively. We discuss the limitations of our approach in Section 7.

**LoRDO Communication and Memory Savings.** Compared to `DDP` with low-rank optimizers, using `LoRDO` re-

alizes a communication benefit of $(\frac{1+\frac{r}{q}}{K_x} + \frac{1}{K_u} + \frac{1}{K_v})^{-1}$ due to its infrequent communication (Iacob et al., 2026b). In the case of DDP with full-rank Adam, this reduction improves to $(\frac{1+\frac{r}{q}}{K_x} + \frac{r}{K_u \cdot p} + \frac{r}{K_v \cdot p})^{-1}$ due to the low-rank optimizer states, in addition to the lower optimizer state memory overhead of $O(\frac{p}{r})$. For communication-efficient training methods using Adam locally, LoRDO-Global reduces communication and memory overhead by $\frac{3pq}{pq+pr+2rq}$. Transmitting the global projection matrix to workers penalizes LoRDO-Global relative to LoRDO-Local. Yet, this overhead is negligible given $r \ll p, q$ in the regimes for which LoRDO is designed. We provide a more detailed discussion of these points in Sections A.5 and A.6.

## 4. Experimental Framework

Building on our theoretical motivations in Section 3, we investigate the following research questions:

**RQ1** Do global projections stagnate learning, as predicted?

**RQ2** Do full-rank quasi-hyperbolic momentum terms alleviate stagnation, as predicted?

**RQ3** Does LoRDO-Global benefit from a larger effective batch size, as per Section 3?

**RQ4** What is the dependence between the synchronization and rank?

**RQ5** How does LoRDO perform against DDP at scale?

### 4.1. Setup

**Models and Data.** Our experiments utilize peri-norm (Kim et al., 2025) decoder-only transformers scaled to 16M, 125M, and 720M parameters, as detailed in Table 4. The 16M variant is used for tuning various hyperparameters (see Section F) and qualitative analysis; the 125M and 720M variants are dedicated to investigating scaling behavior, and for baseline comparisons. We train all models on the SmolLM2 data mixture (Allal et al., 2025) using a sequence length of 2048. For further details, see Section C.

**Optimizers and Tuning Methodology.** Our methods are inspired by the GaLore and LDAdam (Zhao et al., 2024; Robert et al., 2025), initially designed as a low-rank counterpart to Adam (Kingma & Ba, 2015; Loshchilov & Hutter, 2019). Unless otherwise stated, all low-rank methods implement error-feedback (Seide et al., 2014) locally; see more details in Section D.1. For non-QHM experiments, we use $\beta_1 = 0.9, \beta_2 = 0.999$ as recommended by Semenov et al. (2025). For the QHM experiments, we independently tune the optimal $\omega$'s and learning rates $\eta$ for LoRDO-Global and LoRDO-Local, fixing $\beta_1 = 0.999$. Unless otherwise stated, LoRDO-Global always uses full-rank QHM term, where LoRDO-Local uses the low rank form motivated by Section F. Additionally, we leverage the CompleteP

parametrization to transfer the optimal learning rate from the 16M model to our larger models. The DDP baselines independently tune their $\omega, \eta$ parameters. For more details, see Section F

**Baselines.** We compare LoRDO-Global and LoRDO-Local against: i) a DDP analogue using GaLore as the optimizer with LDAdam-style momenta rotations at various ranks, guaranteeing projection matrix update frequency is consistent across DDP and LoRDO, and ii) the full-rank counterparts using Adam for both the DDP setting and the communication-efficient setting. For all infrequent communication training methods, we use the stateful, and provably convergent, approaches of Local Adam (Cheng & Glasgow, 2025) for non-quasi-hyperbolic experiments, and MT-DAO for quasi-hyperbolic experiments, where we set $K = K_x = K_u = K_v = 32$ by default. This extends decoupled sync frequencies as Iacob et al. (2026b); however, this is left for future work. We evaluate ML performance for communication-efficient methods under the same fixed synchronization frequency following prior work (Charles et al., 2025). We split the dataset in an IID fashion across 4 workers using $1\times$ H100 per worker.

**Metrics** Our primary evaluation metric is the mean perplexity across workers. Additionally, we measure how similar the bases of two consecutive rotation matrices $Q_t$ and $Q_{t-1}$ using the Mean Squared Singular Value $MSSV = \frac{1}{r}\sum_{i=1}^{r}\sigma_i^2$ of the rotated basis matrix $R_t = Q_t^\top Q_{t-1}$. We also report the $r^{th}$ spectral gap of a matrix $U$, which is the difference $\sigma_r - \sigma_{r+1}$ of the eigenvalues and we report the stable rank of a matrix U where $sr(U) = \|U\|_F^2 / \|U\|_2^2$. To ensure clarity when evaluating communication or memory efficiency, we *explicitly specify the exact object of reference*. This allows us to differentiating between savings isolated strictly to the *optimizer states* and the *total system payload*, which additionally accounts for the pseudo-gradient and projection matrix overheads.

## 5. Evaluation and Discussion

In this section, we provide a detailed evaluation of LoRDO, where we validate our theoretical findings, namely that i) global projections are superior only when full subspace exploration is enabled (Sections 5.1 to 5.3), ii) lower ranks are more sensitive to delayed communication (Section 5.4) and iii) that near-parity with DDP is reached at scale (Section 5.5). We defer all additional ablations to Section D.

### 5.1. Naïve Global Projections Stagnate Learning (RQ1)

We begin our evaluation by analyzing whether a global projection matrix, without a full-rank quasi-hyperbolic momentum term, stagnates learning, as shown in Figure 1a. Validating our derivations in Section 3, ensuring a global

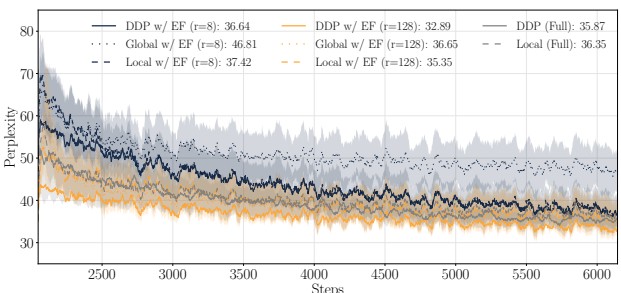

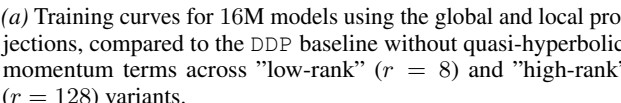

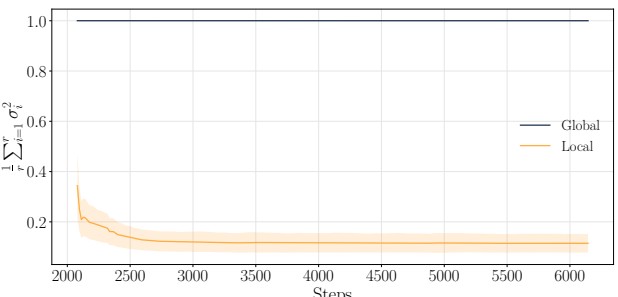

*(a)* Training curves for 16M models using the global and local projections, compared to the DDP baseline without quasi-hyperbolic momentum terms across "low-rank" ($r = 8$) and "high-rank" ($r = 128$) variants.

*(b)* MSSV of the rotation matrix $R_t = Q_t^\top Q_{t-1}$ for the global and local projections. While the LoRDO-Local is able to effectively refresh its projection matrices and explore the full solution space, LoRDO-Global remains stagnant.

*Figure 1.* Global projection matrix pathologies. LoRDO-Global fails to learn when quasi-hyperbolic momentum terms have not been applied due to the projection bases failing to update throughout the duration of local training.

projection is superior in the first few steps of training, following the warmup period, as it uses a projection matrix that is obtained from higher quality pseudo-gradients relative to the per-worker-generated projection matrix. However, the model fails to maintain this improvement as it is unable to update its $r$-rank subspace (Figure 1b), as predicted. LoRDO-Local, instead, explores the full solution space by refreshing its projection matrices throughout the duration of training (Figure 1b). Additionally, we see that LoRDO-Local maintains the same regularizing properties as seen in Robert et al. (2025), where methods of lower rank are able to match or outperform their high-rank variants.

> **Stagnated Subspace:** Without full-rank signals, the LoRDO-Global stagnates learning; all workers optimize within a specific $r-$rank subspace determined by the first server's projection matrix for the duration of training.

### 5.2. LoRDO Enables Full Exploration (RQ2)

In Figure 2a, we show that injecting a full-rank quasi-hyperbolic momentum term alleviates learning stagnation, allowing for full subspace exploration. Across lower ranks ($r \in \{8, 16, 32, 64\}$)), LoRDO-Global consistently outperforms its local variant, and more readily matches the performance of the DDP counterpart, where LoRDO-Global and LoRDO-Local recover MT-DAO (Iacob et al., 2026a) when a full-rank representation is reached. To determine the cause of this performance result, we focus on the stable rank and the signal used to determine the projection, and the spectral gap of the resulting projection, in Figure 3.

Figure 3 shows that while the spectral gap of the two methods decreases as the projection rank increases, as is expected due to the power assumption (Xie et al., 2023), the spectral gap of the global projection, which leverages the aggregated pseudo-gradient, is orders of magnitude larger than the local

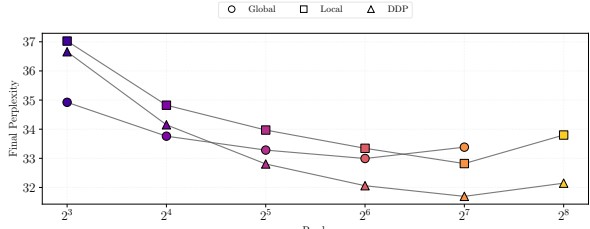

*(a)* Final perplexity for 16M models trained with LoRDO variants compared to DDP.

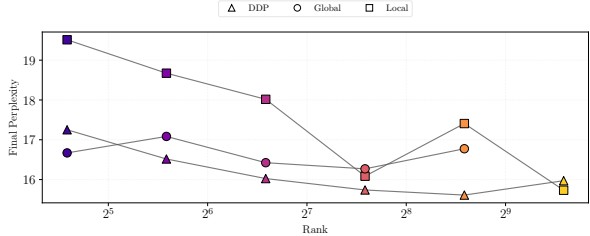

*(b)* Final perplexity for 125M models trained with LoRDO variants compared to DDP.

*Figure 2.* LoRDO with global projections offers superior resilience to small-batch regimes compared to the local projection method. Particularly under heavy memory constraints, which necessitate low ranks, LoRDO-Global surpasses DDP.

counterpart. Observing full derivation of the matrix instability $\Delta(\hat{Q})$ in Section A.10, an increase in the stable rank is inversely proportional to the instability of the projection matrix $\Delta(\hat{Q}) \propto \frac{1}{\delta_r}$. This supports our derivations in Section 3, which show that using a local projection matrix is a suboptimal choice. Furthermore, unlike LoRDO-Local, whose stable rank remains consistent across projection ranks, the LoRDO-Global displays an inverse dependence across projection ranks. We argue that, since the pseudo-gradient has a richer history (including curvature information) for

use in SVD, it is better able to identify principled directions that are more beneficial for the global optimization procedure. However, this comes at the cost of increased noise sensitivity as $r$ increases, leading to greater instability. We provide a more detailed ablation in Section D.8.

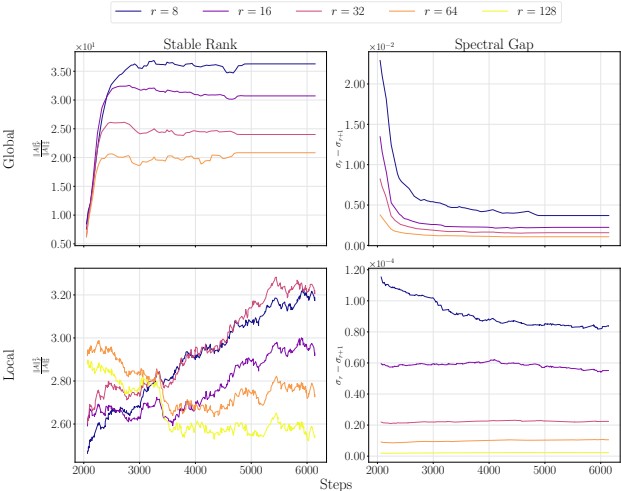

*Figure 3.* Stable rank (left column) and spectral gap (right column) for the attention layers on the 16M parameter model for `LoRDO-Global` (top row) and `LoRDO-Local` (bottom row) of `LoRDO`.

> **`LoRDO` Improves Stability and Performance:** When applying a full-rank quasi-hyperbolic momentum, the `LoRDO-Global` is able to benefit from i) unifying the bases across workers and ii) providing a higher quality projection matrix. This results in an improved optimization trajectory relative to `LoRDO-Local`, especially at lower ranks.

### 5.3. `LoRDO-Global` Improves Projection Quality (RQ3)

In Figure 4, we present an ablation across the number of workers and the batch size used in our 16M parameter experiments, where $r = 8$, to investigate whether the stability of the projection matrix depends on the effective batch size, as we identified in Section 3. Throughout, the global batch size $|B_G|$ is fixed to 64, and we alter the number of workers $|M|$ and the local batch size $|B|$ to maintain $|B_G|$. As predicted, `LoRDO-Global` is much less sensitive to changes in the number of workers than the local variant, because its projection is based on the pseudo-gradient aggregate across workers, with a batch size of $|MB|$. Given that `LoRDO-Local` uses gradients with a batch size $B$, it quickly becomes destabilized as it enters a noise-dominated regime when $B$ decreases. Additionally, these findings are consistent even when the $|B_G|$ is increased beyond its critical batch size for a given model size. In Figure 5, we con-

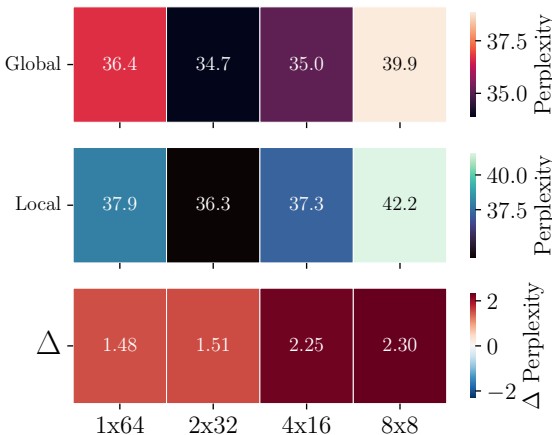

*Figure 4.* Ablation across number of workers and local batch size ($M \times B$) for 16M parameter experiments where $r = 8$. The global batch size (or effective batch size) is 64. We present this ablation for both `LoRDO-Global` and Local and the difference $\Delta = PPX_{\text{Local}} - PPX_{\text{Global}}$. As predicted, we find that `LoRDO-Local` is more sensitive to changes in the local batch size. We provide a larger comparison across varying $K$ in Figure 15.

duct an iso-token comparison between `LoRDO-Global` and `LoRDO` with a $2\times$ larger batch size and a fixed worker size $M$ across varying ranks for 125M models. Despite the improved signal (due to a larger local gradient) for `LoRDO-Local`, `LoRDO-Global` still maintains its superiority (particularly at small ranks as before) due to using a larger effective batch size to determine the projection matrix. `LoRDO-Global` also improves performance relative to the `LoRDO-Local` even when its global batch size is kept constant, further reinforcing the benefit of using a globally, and not locally, computed projection matrix.

> **Global Projections Improve Stability:** Since the stability of the projection matrix scales with $O(\frac{1}{\sqrt{B}})$ (see Section 3), using the global pseudo-gradient improves the projection stability as the worker batch size decreases.

### 5.4. Lower Ranks Are Less Tolerant To More Infrequent Synchronization (RQ4)

We now investigate the interaction between the synchronization interval, where we fix $K_x = K_u = K_v = K$ for simplicity, and the rank chosen for `LoRDO`. Following previous work (Iacob et al., 2026b; Douillard et al., 2023), we consider $K \in \{32, 64, 128, 256, 512, 1024\}$ to map the performance degradation boundaries of `LoRDO` while highlighting the regimes where its relative efficiency gains over `DDP` are most pronounced. In Figure 6, we present this ablation for a "low-" and "high-rank" setting of `LoRDO-Global` and `LoRDO-Local` with/without the quasi-hyperbolic term. Irrespective of the quasi-hyperbolic

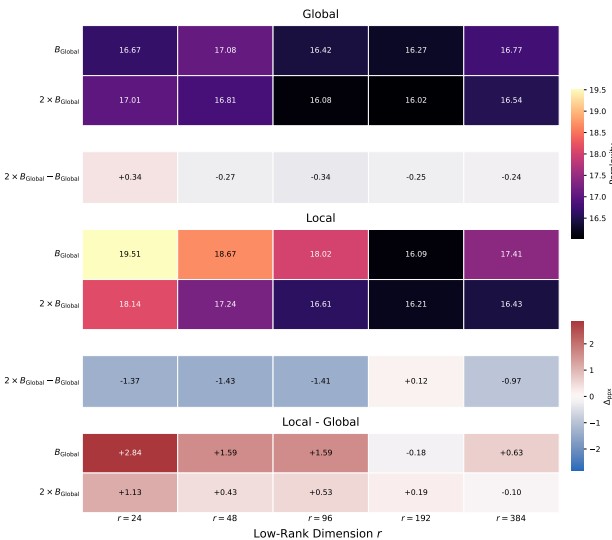

*Figure 5.* Comparison of `LoRDO-Global` and `LoRDO-Local` for 125M models across ranks for $B_{\text{Global}} = 256$ and $2 \times B_{\text{Global}} = 512$ settings. $M = 4$ in this experiment. We find that the effect increasing the global batch size is more pronounced for `LoRDO-Local`. However, `LoRDO-Global` still maintains its improved performance over `LoRDO-Local`, where this is done while maintaining $B_{\text{Global}} = 256$ in certain cases.

term, we find that lower ranks ($r = 8$) are more sensitive to decreases in the synchronization frequency. Specifically, as the synchronization frequency is lowered (i.e., higher $K$), lower-rank variants are more likely to deviate from their high-synchronization counterparts and potentially diverge during training. In higher-rank regimes ($r = 128$), there is less difference across the synchronization frequencies. We also note that applying quasi-hyperbolic momentum terms reduces the extent to which this affects performance, although rank-level sensitivity persists. We posit this is due to the higher $\beta_1$ offering a longer half-life to the first momentum term, allowing it to be synchronized less often (see Iacob et al. (2026a)).

> **Low Ranks Are Sensitive to Sync Freq:** Lower ranks are more sensitive to infrequent synchronization. Quasi-hyperbolic momentum mitigates this instability, maintaining performance while reducing communication overhead.

### 5.5. LoRDO Maintains Benefit at Scale (RQ5)

To determine `LoRDO`'s practical benefit at scale, we use it to train the 125M and 720M scales as seen in Figure 2b and Table 1, respectively. In the case of the 720M results, we select $r = 256$ such that this provides an $8\times$ improvement for the optimizer state overhead relative to the full $r = 2048$ counterpart. We complement our perplexity results with a downstream task evaluation, reporting per-task and average task performance. We present results across the full training

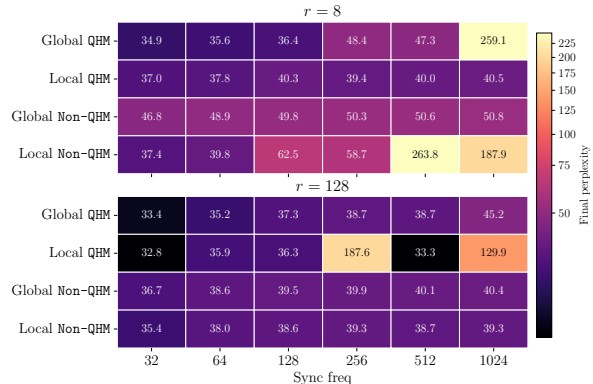

*Figure 6.* Ablation across synchronization frequency for `LoRDO` variants and QHM terms for 16M parameter models. Lower ranks are more sensitive to delays in synchronization. In addition to offering more stable performance, QHM terms reduce this sensitivity with an increased $\beta_1$.

duration for the 720M model in Figure 17.

Observing Figure 2b, `LoRDO-Global` maintains its superior performance over `LoRDO-Local`, where this is the most pronounced in lower rank regimes, as in the 16M case. Scaling to 720M parameters, we find that the benefit of `LoRDO` becomes effectively indistinguishable from its low-rank `DDP` counterpart. Specifically, `LoRDO` exhibits a reduction in perplexity of less than $1\%$, providing the same performance in downstream benchmarks, while achieving a communication reduction of $10\times$. `LoRDO-Global` maintains its benefits for communication-efficient training by reducing the communication overhead of full-rank `DDP` by $\approx 25\times$ and the memory overhead for the optimizer states by $8\times$. Additionally `LoRDO-Global` reduces the memory and communication overhead for optimizer states by $8\times$.

*Table 1.* 720M results comparison across final perplexity and downstream task accuracy. `LoRDO` matches its low-rank `DDP` counterparts, while substantially reducing communication.

| Metric | | $r = 256$ | | $r = 2048$ | |
| --- | --- | --- | --- | --- | --- |
| | DDP | LoRDO-Local | LoRDO-Global | MT-DAO | DDP |
| ARC-Challenge (0-shot) | 28.8 | 29.2 | 30.5 | 31.3 | 31.0 |
| ARC-Easy (0-shot) | 55.5 | 55.1 | 54.7 | 56.6 | 57.2 |
| HellaSwag (0-shot) | 40.3 | 40.0 | 40.1 | 42.2 | 43.2 |
| MMLU (5-shot) | 31.1 | 30.6 | 31.0 | 31.3 | 31.3 |
| PIQA (0-shot) | 68.6 | 68.6 | 67.1 | 68.3 | 69.2 |
| **Avg. All Tasks** | 44.8 | 44.7 | 44.7 | 45.9 | 46.4 |
| Perplexity | 10.34 | 10.56 | 10.41 | 9.98 | 9.85 |

> **LoRDO is Performant at Scale:** `LoRDO` provides near-parity with its low-rank `DDP` counterpart across scales, in both perplexity and downstream task performance, showing that competitive performance can be achieved with significantly lower communication overhead. Furthermore, it remains competitive with the full-rank methods despite an $8\times$ reduction in memory and communication costs.

## 6. Related Work

**Distributed Training with Infrequent Communication.** Prior research has focused on local update methods to mitigate the communication bottlenecks of standard Distributed Data Parallelism (DDP). Local SGD (Stich, 2019) enables workers to perform multiple local steps before averaging parameters. In LLM pre-training, DiLoCo (Charles et al., 2025) achieved high performance by combining local updates with Nesterov momentum as an outer optimizer. Local Adam (Cheng & Glasgow, 2025) proved convergence for adaptive optimizers by synchronizing all states, albeit tripling communication costs. DES-LOC (Iacob et al., 2026b) improved this by decoupling parameter and momenta synchronization, with the extension by Iacob et al. (2026a) to ensure that the momenta half-lives matched their synchornization intervals. Our work builds on this multi-timescale framework, primarily targeting the memory constraints of full-rank optimizer states.

**Optimization via Low-Rank Gradient Projection.** Unlike LoRA (Hu et al., 2022), which restricts optimization to adapters, GaLore (Zhao et al., 2024) projects full-rank gradients into a low-rank subspace via SVD, reducing memory without altering training dynamics. LDAdam (Robert et al., 2025) enhances this with Block Power Iteration and error feedback, while Dion (Ahn et al., 2025) uses rank-$r$ orthogonalization to reduce overhead. These methods reduce memory in synchronous settings but have not been adapted for infrequent synchronization.

**Distinction from Communication Compression Methods.** A parallel line of work reduces communication volume via payload compression techniques such as quantization (Alistarh et al., 2017), sparsification (Lin et al., 2018), or mixes thereof, as seen in CocktailSGD (Wang et al., 2023), DiLoCoX (Qi et al., 2025), and SparseLoCo (Sarfi et al., 2025). We distinguish our approach from these methods on three critical grounds. First, these methods compress the pseudo-gradient only for transmission and do not reduce local memory costs. Second, they perform compression after local training, which can degrade model quality; conversely, optimization directly in a low-rank subspace has been shown to match or improve performance due to implicit regularization effects (Robert et al., 2025), a finding corroborated by our empirical results, while providing the communication benefits as a byproduct. Third, purely compression-based methods do not account for the transmission of optimizer states, which is known to be necessary for convergence guarantees and stability (Cheng & Glasgow, 2025). While our method does not inherently require state transmission, its intrinsic low-rank structure offers a principled compression mechanism. This reduces communication costs in proportion to the rank reduction, making the transmission of states more practical. Section E presents an extended overview.

## 7. Limitations and Future Work

Our empirical validation at the 720M scale is limited; computational constraints prevented an analysis of trends across varying ranks, longer training durations, and further extensions to larger model sizes. However, we posit that the trends observed at the 16M and 125M scales will hold. Secondly, although we present theoretical justifications for our design decisions in LoRDO, we do not provide a formal convergence proof for LoRDO. Nevertheless, we leave this for future work as we believe the theory should follow from prior theoretical results. Specially, this would involve constructing a virtual iterate sequence to handle the multi-step quasi-hyperbolic momentum updates as per Iacob et al. (2026a), after which bounding the remaining client and virtual drift terms becomes a straightforward application of the error feedback machinery developed in Robert et al. (2025).

In Section 3, although we hypothesize that LoRDO-Local causes interference when aggregating pseudo-gradients in Non-IID settings, we do not conduct experiments to validate this claim, and is left as a direction for future work. Additionally, we note that LoRDO's experimental framework focuses on low-rank optimizers developed for adaptive optimizers such as Adam (Kingma & Ba, 2015). A natural extension is to incorporate matrix-based optimizers like Muon (Jordan et al., 2024), as LoRDO mainly tackles the object used to compute the projection matrix. Specifically, this could be achieved by either: i) additionally down-projecting the gradient prior to momentum calculation, or ii) using a low-rank calculation similar to Dion (Ahn et al., 2025) where LoRDO would serve as the communication-efficient training generalization. We leave full empirical investigation of this for future work.

## 8. Conclusion

LoRDO resolves the tension between projection stability and subspace exploration in low-rank communication-efficient optimization. By using an aggregated pseudo-gradient with full-rank quasi-hyperbolic momentum terms incorporated in the local update, we eliminate subspace stagnation without incurring the high variance of local methods. Furthermore, this principled design allows for LoRDO to match the performance of synchronous low-rank DDP at scale while communicating $\approx 25\times$ less. Furthermore, it reduces optimizer memory and communication costs by up to $12\times$ compared to previous principled communication-efficient adaptive optimizers. Crucially, LoRDO exhibits superior resilience in memory-constrained settings, making pre-training feasible on hardware with limited resources. Ultimately, LoRDO extends model training beyond single data centers, paving the way for efficient learning in decentralized and memory and bandwidth-constrained environments.

## Acknowledgments

The authors thank the anonymous reviewers for their valuable feedback which has improved the quality of this work. This research was supported by the following entities: The Royal Academy of Engineering via DANTE (a RAEng Chair); the European Research Council, specifically the REDIAL project; SPRIND under the composite learning challenge; Google through a Google Academic Research Award; in addition to the Ministry of Education of Romania (through the Credit and Scholarship Agency). MS was supported by Research England under the Expanding Excellence in England (E3) funding stream, which was awarded to MARS: Mathematics for AI in Real-world Systems in the School of Mathematical Sciences at Lancaster University.

## Impact Statement

This paper presents work whose goal is to advance the field of machine learning. There are many potential societal consequences of our work, none of which we feel must be specifically highlighted here.

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

# Appendix

## Table of Contents

# A. Extended Mathematical Derivations

## A.1. Unrolled Low-Rank Updates Through Time

Below, we provide a full mathematical derivation of the local optimization procedure for a worker $m$ to support the arguments presented in Section 3. Initially, we consider the effect of local optimization without any quasi-hyperbolic momentum terms.

We follow the setting of `DES-LOC` (Iacob et al., 2026b) and `Local Adam` (Cheng & Glasgow, 2025) (both of which encompass the typical `FedOpt`(Reddi et al., 2021) setting). At first, we consider the setting where we set the number of local steps $K$ to one. This gives the following local update computation when using a low-rank adaptive optimizer:

$$\theta_{t+1}^m \leftarrow \theta_t - \eta_t^m Q_t^m \alpha_t^m \tag{2}$$

where $\theta_t$ is the model received by the worker at the previous synchronization boundary. Computing the per-worker pseudo-gradients:

$$\Delta_t^m \leftarrow \theta_t - \theta_{t+1}^m \tag{3}$$
$$\Delta_t^m \leftarrow \theta_t - (\theta_t - \eta_t^m Q_t^m \alpha_t^m) \tag{4}$$
$$\Delta_t^m \leftarrow \eta_t^m Q_t^m \alpha_t^m \tag{5}$$

Now, we increase the number of local steps to some generic amount $K$ to construct similar decompositions, albeit unrolled through time.

$$\Delta_t^m \leftarrow \theta_{t-K} - \theta_t^m \tag{6}$$
$$\Delta_t^m \leftarrow \theta_{t-K} - (\theta_{t-1}^m - \eta_{t-1}^m Q_{t-1}^m \alpha_{t-1}^m) \tag{7}$$
$$\Delta_t^m \leftarrow \theta_{t-K} - (\theta_{t-2}^m - \eta_{t-2}^m Q_{t-2}^m \alpha_{t-2}^m - \eta_{t-1}^m Q_{t-1}^m \alpha_{t-1}^m) \tag{8}$$

$$\Delta_t^m \leftarrow \theta_{t-K} - (\theta_{t-K} - \sum_{\tau=t-K}^{t} \eta_\tau^m Q_\tau^m \alpha_\tau^m) \tag{9}$$

$$\Delta_t^m \leftarrow \sum_{\tau=t-K}^{t} \eta_\tau^m Q_\tau^m \alpha_\tau^m \tag{10}$$

## A.2. Aggregated Pseudo-gradient without Quasi-hyperbolic Momentum Terms

As before, we ignore the learning rate. However, in this case, we have assumed that the projection matrix gets updates at every step, causing a dependence on the time axis, namely:

$$\sum_{\tau=t-K}^{t} Q_\tau^m \alpha_\tau^m \neq \sum_{\tau=t-K}^{t} Q_\tau^m \sum_{\tau=t-K}^{t} \alpha_\tau^m \tag{11}$$

However, if we instead assume, for example, that $Q^m$ remains fixed for the entire local training trajectory (as is configured with `GaLore` Zhao et al. (2024) for each worker in our experiments), or if each worker $m$ receives a global projection matrix $Q$, we can derive the following update as the low rank projection matrix no longer depends on the local step:

$$\Delta_t^m \leftarrow Q^m \sum_{\tau=t-K}^{t} \eta_\tau^m \alpha_\tau^m \text{ or } \Delta_t^m \leftarrow Q \sum_{\tau=t-K}^{t} \eta_\tau^m \alpha_\tau^m \tag{12}$$

When viewed from the perspective of the global optimization step occurring on the outer optimizer:

$$\theta_{t+1} \leftarrow \theta_t - \eta_t^s \Delta_t \tag{13}$$

$$\theta_{t+1} \leftarrow \theta_t - \eta_t^s \frac{1}{|M|} \sum_{m=1}^{M} \Delta_t^m \tag{14}$$

$$\theta_{t+1} \leftarrow \theta_t - \eta_t^s \frac{1}{|M|} \sum_{m=1}^{M} \sum_{\tau=t-K}^{t} \eta_\tau^m Q_\tau^m \alpha_\tau^m \tag{15}$$

In the case of using a fixed local projection matrix, as is the case for `LoRDO-Local`:

$$\theta_{t+1} \leftarrow \theta_t - \eta_t^s \underbrace{\frac{1}{|M|} \sum_{m=1}^{M} Q^m \sum_{\tau=t-K}^{t} \eta_\tau^m \alpha_\tau^m}_{\text{Pseudo-gradient signal}} \tag{16}$$

This modifies slightly in the case of ensuring a global projection matrix $Q_t$ as is the case for `LoRDO-Global`.

$$\theta_{t+1} \leftarrow \theta_t - \eta_t^s \underbrace{\frac{1}{|M|} Q_t \sum_{m=1}^{M} \sum_{\tau=t-K}^{t} \eta_\tau^m \alpha_\tau^m}_{\text{Pseudo-gradient signal}} \tag{17}$$

### A.3. Discussions on Truncated Rank Representations

As discussed in Section 3, there are two solutions that emerge to determine the worker's projection matrix: a *global* variant where each worker receives the same unified projection matrix, and a *local* variant where each worker determines its own projection matrix. While the global projections provide a more principled unification of the workers, and a higher quality projection matrix estimation, this introduces a subspace stagnation in the global optimization direction following the server accumulation without any full-rank quasi-hyperbolic momentum. Specifically, we present the aggregation step when accumulating the pseudo-gradient from $M$ workers for the local (left, assuming the local projection matrix remains stagnant throughout the local training procedure) and global (right) variants (isolating the pseudo-gradient signal from Equations (16) and (17):

$$\Delta_t \leftarrow \frac{1}{|M|} \sum_{m=1}^{M} \sum_{\tau=t-K}^{t} \eta_\tau^m Q_t^m \alpha_\tau^m \tag{18}$$

$$\Delta_t \leftarrow \frac{1}{|M|} \sum_{m=1}^{M} Q^m \sum_{\tau=t-K}^{t} \eta_\tau^m \alpha_\tau^m \tag{19}$$

$$\Delta_t \leftarrow \frac{1}{|M|} \sum_{m=1}^{M} \sum_{\tau=t-K}^{t} \eta_\tau^m Q_t^m \alpha_\tau^m \tag{20}$$

$$\Delta_t \leftarrow \frac{1}{|M|} Q \sum_{m=1}^{M} \sum_{\tau=t-K}^{t} \eta_\tau^m \alpha_\tau^m \tag{21}$$

In the case of the local model, as derived in Section A.1, we see that the projection matrix $Q$ has no time dependence as we constrain it to remain constant for a window of $K$ steps. However, each client has computed its own $r$-rank projection matrix, capturing the benefit of its local data distributions, enforcing the dependence across the worker aggregation. When aggregating the signal, although we sum $M$ individual $p \times q$ matrices of rank $r$ (assuming the rank of the pseudogradient itself is rank $r$), the rank of the overall pseudo-gradient signal $\Delta_t$ scales with the number of workers. Given sufficiently many workers, the aggregation can recover a full-rank representation, up to $\min(p, q)$. This occurs because the individual matrices may span different, or even orthogonal, subspaces within the original $p \times q$ space; consequently, their sum can recover the full space (Horn & Johnson, 1985). Furthermore, there is a temporal component: even if the aggregated pseudo-gradient remains rank-deficient at a single step (i.e., $M \times r < \min(p, q)$), the iterative optimization process accumulates enough distinct subspaces over time to eventually span the full space. This mechanism is leveraged by Zhao et al. (2024); Robert et al. (2025).

The global projection, however, is independent across the time and worker axes as all workers share the same projection matrix $Q$. As such, when aggregating the signal, and then returning the projection to the full-rank basis, the pseudo-gradient signal is at best a rank $r$ matrix; there was no diversity contributed from the individual data distributions available to each client. Consequently, when we compute the next projection matrix $Q_{t+1}$ for `LoRDO-Global` through an SVD operation, the new projection lies entirely within the subspace spanned by the old projection matrix. The $r$ bases of the previous projection matrix are indeed the top $r$ most principled vectors of this new full dimensional space. As such, the projection matrix will never "refresh" at the synchronization interval. This does not happen in the local method as the workers each compute their own projection matrices on their local gradient signal. We visualize this pathology in Figure 1.

**A.4. The Impact of Quasi-hyperbolic Momentum Terms on Learning and on Up-Link Communication**

In Section 3.2, we introduced a novel contribution where we adapt quasi-hyperbolic momentum terms for low-rank optimizers. Specifically, we show that quasi-hyperbolic momentum terms can be introduced either in a low- or full-rank form. Below, we elucidate the impact this has on both the representation of low-rank optimization and the subsequent communication benefits that can be realized. We adapt the derivations to present the low- and full-rank quasi-hyperbolic momentum variants on the left and right, respectively, without making assumptions on the structure of the projection matrix.

$$\Delta_t \leftarrow \frac{1}{|M|} \sum_{m=1}^{K} \sum_{\tau=t-K}^{t} \eta_\tau^m Q_t^m \alpha_\tau^m \qquad\qquad \Delta_t \leftarrow \frac{1}{|M|} \sum_{m=1}^{M} \sum_{\tau=t-K}^{t} \eta_\tau^m Q_t^m \alpha_\tau^m$$

$$\Delta_t \leftarrow \frac{1}{|M|} \sum_{m=1}^{K} \sum_{\tau=t-K}^{t} \eta_\tau^m Q_t^m \left[ \frac{\omega u_\tau^m + (1-\omega)\hat{g}_\tau^m}{\sqrt{v_\tau^m} + \epsilon} \right] \qquad \Delta_t \leftarrow \frac{1}{|M|} \sum_{m=1}^{M} \sum_{\tau=t-K}^{t} \eta_\tau^m \left( (1-\omega)\frac{\hat{G}_\tau^m}{\mu(\sqrt{\hat{v}_\tau^m} + \epsilon)} + \omega Q_\tau^m \left[ \frac{u_\tau^m}{\sqrt{v_\tau^m} + \epsilon} \right] \right)$$

Observing the structure of the low-rank quasi-hyperbolic momentum variant, we see that the additional signal is still constrained to the low-rank subspace determined by the projection matrix. In the case of LoRDO-Global, adding the low-rank quasi-hyperbolic momentum term does not alleviate the stagnated learning pathology we observed in Section 3.

In the case of the full-rank quasi-hyperbolic term, we see that the signal becomes full-rank on each worker due to the injection of the full-rank gradient. This allows the learning dynamics to explore diverse subspaces overtime as per the previosu section. As such, irrespective of whether one uses the local or global variant of LoRDO, full subspace learning is guaranteed as the aggregated pseudo-gradient will always be full rank provided that the correct quasi-hyperbolic momentum form is applied.

As mentioned in Section 3, we note here that the induction of the quasi-hyperbolic momentum has a dual use. While the primary benefit is to enable full-subspace explorationm it simultaneously allows the momentum half-lives to safely match their synchronization intervals, consistent with Iacob et al. (2026a). This leads to improved empirical performance for both LoRDO-Global and LoRDO-Local, corroborating prior work.

**A.5. Addressing Communication**

In this section, we provide a discussion into the communication overhead of LoRDO. Table 2 summarises the per-payload communication amount across both DDP and the communication-efficient training regimes.

A.5.1. WORKER COMMUNICATION

Observing the above equations for local and LoRDO-Global, we can realize an additional communication benefit on the model parameters (or pseudo-gradient). Instead of communicating the entire full-rank pseudo-gradient signal, we alleviate the communication overhead by decomposing the pseudo-gradient into its low-rank projection matrix and an accumulated update buffer across time. Notice that although the outer optimizer receives the two low-rank constructions, we are able to rebuild the full-rank pseudo-gradient signal as if this were rebuilt locally on each worker. In the case of a globally unified projection matrix, the communication cost reduces to just the accumulated low-rank pseudogradient through time as the server already stores a copy of the global projection matrix it sent to each worker. This derivation holds for all server-side aggregation strategies, not just averaging as we have used for the purposes of our experiments.

In the case of applying quasi-hyperbolic momentum terms, the same communication benefit can be realized should the quasi-hyperbolic momentum terms be applied in their low-rank form. However, if they are applied in the full-rank form, one can no longer decompose the pseudo-gradient signal. As such, the communication cost of this regime reverts to the same as communicating dense pseudo-gradients in the case of previous methods (Iacob et al., 2026a;b).

However, in these methods, not only are the parameters (or pseudo-gradients) communicated, but a similar operation is performed to synchronize the optimizer states which ensures provable convergence. In this regard, LoRDO, which applies low-rank optimizers locally with a global projection matrix, also affords communication reduction as this $rq$ low-rank structure is communicated instead of the $pq$ matrix for each optimizer state.

A.5.2. ADDRESSING DOWN-LINK COMMUNICATION

Observing the construction presented in Equation (16), downlink communication for the local method is always guaranteed to be full rank. Given that each worker's projection matrix $Q^m$ is non-identical, this element cannot be pulled out of the second

summation, which would allow us to communicate two low rank structures as we did with the up-link communication. In the case of the global variant, in the case of no global quasi-hyperbolic momentum, a reduction in down-link communication can be achieved. Assuming that each worker maintains the previous projection matrix $Q_t$ and previously received model $\theta_t$, each worker only needs to receive the aggregated low rank pseudo-gradient signal $\sum_{m=1}^{M} \sum_{\tau=t-K}^{t} \eta_\tau^m \alpha_\tau^m$ to compute the new model $\theta_{t+1}$ locally. In addition to this, each worker needs to receive the new globally computed projection matrix $Q_{t+1}$. In the case of `LoRDO-Global` with full-rank quasi-hyperbolic momentum terms, the pseudo-gradient update signal is indeed full rank, and thus no additional communication benefit can be achieved. Instead, `LoRDO-Global` incurs an additional $O(pr)$ cost to transmit the projection bases to each worker. However, given the fact that $r \ll p, q$, this additional communication cost is negligible, and is counteracted by the additional overhead afforded by synchronizing the momenta states in their low-rank forms.

*Table 2.* Communication Cost Comparison on a per-payload basis between `DDP` and `LoRDO-Global`/`LoRDO-Local`. We assume that the pseudo-gradient and momenta parameters are communicated at the same interval $K$

| Method | QHM | Up-Link | Down-Link |
|---|---|---|---|
| Global | N/A | $O(3rq)$ | $O(pr + 3rq)$ |
| | Low | $O(3rq)$ | $O(pr + 3rq)$ |
| | Full | $O(pq + 2rq)$ | $O(pq + pr + 2rq)$ |
| Local w/ ( Fixed Projection) | N/A | $O(pr + 3rq)$ | $O(pq + 2rq)$ |
| | Low | $O(pr + 3rq)$ | $O(pq + 2rq)$ |
| | Full | $O(pq + 2rq)$ | $O(pq + 2rq)$ |
| `MT-DAO`/ `DES-LOC` | – | $O(3pq)$ | $O(3pq)$ |
| `DDP` | – | $O(pq)$ | $O(pq)$ |

### A.5.3. UNDERSTANDING THE ROLE OF COMMUNICATION FREQUENCY

The primary communication benefit of training methods that employ local updates is that they reduce the communication frequency relative to `DDP`. As per Iacob et al. (2026b), the amount these regimes benefit from infrequent communication can be quantified by the following ratio:

$$\left(\frac{1}{K_x} + \frac{1}{K_u} + \frac{1}{K_v}\right)^{-1} \tag{22}$$

$$\tag{23}$$

where $K_x, K_u$ and $K_v$ are the update frequencies of the model parameters and optimizer states respectively. Relative to a `DDP` optimizer using full-rank momenta (like `Adam`), we first realize a $\frac{p}{r} \times$ reduction in the memory overhead to transmit the low-rank momenta terms. This leads to an improvement of:

$$\left(\frac{1}{K_x} + \frac{1}{K_u \cdot \left(\frac{p}{r}\right)} + \frac{1}{K_v \cdot \left(\frac{p}{r}\right)}\right)^{-1} \tag{24}$$

in the case of `LoRDO-Local` where projection matrices are not determined globally. In the case of the latter, we incur a slight cost relative to `DDP`. Instead of transmitting only a dense pseudo-gradient (or model parameter) $pq$, we also transmit the global projection matrix of size $pr$ to the workers. This changes the benefit of the `LoRDO-Global` regime to:

$$\left(\frac{1 + \frac{r}{q}}{K_x} + \frac{1}{K_u \cdot \left(\frac{p}{r}\right)} + \frac{1}{K_v \cdot \left(\frac{p}{r}\right)}\right)^{-1} \tag{25}$$

In any case, given that $r \ll p, q$, the additional overhead is negligible. When compared to communication-efficient regimes that use full-rank optimizers, the total benefit is $\frac{3pq}{pq+2rq}$ in the case of `LoRDO-Local` and $\frac{3pq}{pq+pr+2rq}$ in the case of `LoRDO-Global`.

### A.6. **LoRDO** Worker Memory Overhead.

In Table 3, we present the memory overhead complexity for each of the variants of `LoRDO`, relative to the full-rank `Adam` baseline. Across all variants, we see that the consistent trade-off in memory overhead is modulated by the choice of the rank

*Table 3.* Memory overhead for all LoRDO variants compared to full-rank counterparts. ∗ indicates additional memory saving by rewriting the error feedback to the gradient variable as per Robert et al. (2025).

| | Model Parameters | Optimizer States | Projection Matrix | Uplink Time Buffer | Error Buffer | Overhead |
|---|---|---|---|---|---|---|
| Adam | $O(pq)$ | $O(2pq)$ | / | / | / | $O(3pq)$ |
| **Without Uplink Time Buffer** | | | | | | |
| LoRDO Global / Local | $O(pq)$ | $O(2rq)$ | $O(pr)$ | / | $O(pq)$ | $O(pq + pr + 2rq)^*$ |
| **With Uplink Time Buffer** | | | | | | |
| LoRDO Global / Local No QHM and Low-Rank QHM | $O(pq)$ | $O(2rq)$ | $O(pr)$ | $O(rq)$ | $O(pq)$ | $O(pq + pr + 3rq)^*$ |
| LoRDO Global / Local Full-Rank QHM | $O(pq)$ | $O(2rq)$ | $O(pr)$ | / | $O(pq)$ | $O(pq + pr + 2rq)^*$ |

$r$, as expected. Specifically, in cases where $r \ll p, q$, we observe significant memory savings as storing two matrices of $pr$ and $rq$ are considerably less intensive than a $pq$ matrix.

An important characteristic of the communication savings of LoRDO can be observed in Table 3. Specifically, in order to realize the up-link cost savings in Section A.5.1, for the non- and low-rank quasihyperbolic momentum variants, each worker $m$ incurs an additional $O(rq)$ cost to store the accumulated buffer across time between parameter synchronization periods. For full-rank methods, this structure is not possible; as such, it achieves a lower memory overhead traded for an increase in the communication payload size.

Finally, each worker incurs memory overhead to maintain the error-feedback buffer, which is the size of the full-rank gradient. However, as done by Robert et al. (2025), this error feedback buffer can be stored on the full-rank gradient variable for a memory-efficient implementation. Additionally, we note that the low-rank gradient can be deleted right after it has been applied to the first and second moment; as such, it does not incur any extra memory overhead. As such, relative to the full-rank Adam baseline, there is no extra memory overhead in this case.

### A.7. Discussion on the Synchronization of Optimizer States

Following Cheng & Glasgow (2025); Iacob et al. (2026b), we devise LoRDO as a distributed training method that communicates optimizer states by default. As shown in Figure 9, while this increases communication overhead relative to methods such as Douillard et al. (2023); Thérien et al. (2026) that do not communicate optimizer states, synchronizing optimizer states provides significantly greater stability and potentially improved performance compared to disabling it, corroborating prior work (Iacob et al., 2026b).

### A.8. Memory Overhead with Full-Rank Optimizers

Another approach to reducing the memory overhead of the two Adam optimizer states in distributed training is to use a matrix-based optimizer like Muon (Jordan et al., 2024), which maintains only a single optimizer state. With reference to LoRDO, With a memory overhead of $\mathcal{O}(pq + pr + 2rq)$, it automatically reduces memory usage relative to Adam, which requires $\mathcal{O}(3pq)$. However, LoRDO can still provide memory-overhead improvements to Muon ($\mathcal{O}(2pq)$), depending on the chosen rank. Specifically, LoRDO is more efficient when:

$$pq + pr + 2rq < 2pq \tag{26}$$
$$r(p + 2q) < pq \tag{27}$$
$$r < \frac{pq}{(p + 2q)} \tag{28}$$
$$\text{Assuming } p = q, r < \frac{p}{3} \tag{29}$$

Given that $r \ll p, q$ in our experiments (as LoRDO is designed for these settings), this condition is readily met.

### A.9. Discussion on the Choice of Signal for `LoRDO-Global`

In Section 3, we make the design decision to use the projection matrix to compute the global projection matrix $Q_t$. However, in distributed optimization, we also have the opportunity to use the optimizer states as well as the pseudo-gradient signal for this compression operation as they both materialize at the synchronization boundary. However, using the momenta terms instead of, or in concert with, the pseudo-gradient offers no benefit. By definition, the optimizer states are confined to the same subspace as the pseudo-gradient. Furthermore, if you perform SVD on the up-projected momentum signal, the optimal projection matrix that will be computed will be the same that was used in the previous iteration, as in Section A.3.

### A.10. Derivation of Projection Matrix Instability

Below, we provide a more rigorous derivation of the instability formula we had used in Section 3, where we repeat some of the definitions for convenience.

Assume that the stochastic gradient $\hat{G}$ is a perturbed version of the true gradient $G$ such that $\hat{G} = G + E$, where $E$ represents the additive noise incurred through the stochastic sample. Furthermore, we assume that the noise scales proportional to $\frac{\kappa}{\sqrt{B}}$, where $B$ is the batch size and $\kappa$ is the variance of the individual samples (McCandlish et al., 2018; Bottou et al., 2018). Additionally, as shown by Xie et al. (2023), we assume that the singular values of $G$ and $\hat{G}$ follow a power-law where $\sigma_k = Ck^{-\alpha}$ where $\alpha > 0$.

Using the Davis-Kahan $\sin \Theta$ theorem (Stewart & Sun, 1993; Davis & Kahan, 1970), we quantify the stability of the true $r$ rank subspace $Q$ and estimated subspace derived from $\hat{G}$:

$$\| \sin \Theta(Q, \hat{Q}) \|_F \leq \frac{\|E\|_F}{\delta_r}$$

where $\delta_r$ represents the spectral gap $(\sigma_r - \sigma_{r+1})$. We quantify the instability of the projection matrix $\hat{Q}$ as:

$$\Delta(\hat{Q}) \approx \frac{\|E\|_F}{\delta_r}$$

To approximate the spectral gap $\delta_r = \sigma_r - \sigma_{r+1}$, we treat the singular values as a continuous function of the rank index $r$. Given the power-law assumption $\sigma(k) = Ck^{-\alpha}$, the difference between two consecutive singular values corresponds to the magnitude of the gradient of the singular value curve at rank $r$. By applying the first-order Taylor approximation, we write:

$$\delta_r \approx \left| \frac{d\sigma(k)}{dk} \bigg|_{k=r} \right|$$

Differentiating the power-law function with respect to $k$ yields:

$$\frac{d}{dk} \left( Ck^{-\alpha} \right) = -\alpha Ck^{-(\alpha+1)}$$

Taking the absolute value, we obtain the spectral gap approximation $\delta_r \approx \alpha Cr^{-(\alpha+1)}$. Substituting this result and the noise estimate $\|E\|_F \approx \frac{\kappa}{\sqrt{B}}$ back into the instability inequality gives rise to:

$$\Delta(\hat{Q}) \approx \frac{\kappa/\sqrt{B}}{\alpha Cr^{-(\alpha+1)}} = \frac{\kappa}{\alpha C\sqrt{B}} \cdot r^{\alpha+1}$$

# B. Additional Algorithms

---

**Algorithm 2** `LoRDO` Local Variant - Bias Correction Omitted for Ease of Notation

---

**Require: Model tensors, hyper-parameters**

1:     $T, M \in \mathbb{N}_+$ — total optimization steps and number of workers

2:     $\beta_1, \beta_2 \in [0, 1), \omega \in [0, 1]$ — decay rates for each momentum state and QHM convex combination coefficients

3:     $\rho \in \mathbb{R}_+, \{\eta_t\}_{t=0}^{T-1}$ — clipping radius, learning-rate schedule

4:     $K_x, K_u, K_v \in \mathbb{N}_+$ — communication periods for parameters and states

5:     `OuterOpt` $: (\mathbb{R}^d, \mathbb{R}^d) \to \mathbb{R}^d$ — update params using an outer optimizer, averaging by default

6:     `ComputeProjection` $: (\mathbb{R}^{d \times d}, \mathbb{R}) \to \mathbb{R}^{d \times r}$ — Compute projection routine

7:     $x_0^m = x_0 \in \mathbb{R}^d, u_{-1}^m = \mathbf{0}_r, v_{-1}^m = \mathbf{0}_r$ — initial params, first and second momentum

8:     $Q_0 = \mathbb{I}^{d \times r}, E_{-1}^m = \mathbf{0}_{d \times d}, \forall m \in M$ - Identity initial projection matrix and zeroed-out error buffer for each client

**Ensure:** $x_T, u_{T-1}, v_{T-1}$

9:  **for** $t = 0, \dots, T - 1$ **do**

10:      **for all** workers $m = 0, \dots, M - 1$ **in parallel do**

11:         $\hat{G}_t^m \leftarrow \mathbf{clip}(\nabla F(x_t^m; \xi_t^m), \rho)$                Clipped stochastic gradient in full-rank

12:         **if** $((t - 1) \bmod K_x = 0)$ **then**                Update $Q_t^m$ following $K_x$ sync.

13:            $Q_t^m \leftarrow$ `ComputeProjection`$(\hat{G}_t^m + E_{t-1}^m)$     Compute a new global projection matrix with error feedback

14:            $\bar{u}_{t-1}^m \leftarrow Q_t^{m\top} Q_{t-1}^m \bar{u}_{t-1}^m$              Rotate the first moment locally

15:            $\bar{v}_{t-1}^m \leftarrow (1 - \beta_2^t) \left| (Q_t^{m\top} Q_{t-1}^m)^2 (\hat{\bar{v}}_{t-1}^m - (\hat{\bar{u}}_{t-1}^m)^2) + (Q_t^{m\top} Q_{t-1}^m \hat{\bar{u}}_{t-1}^m)^2 \right|$     Rotate the second moment locally

16:         **else**

17:            $Q_t^m = Q_{t-1}^m$                      Maintain stale projection

18:         $\hat{g}_t^m \leftarrow Q_t^{m\top}(\hat{G}_t^m + E_{t-1}^m)$           Low-rank gradient signal with error-feedback

19:         $E_m^t \leftarrow \hat{G}_t^m + E_{t-1}^m - Q_t^m \hat{g}_t^m$            Compute error feedback

20:         $u_t^m \leftarrow \beta_1 \bar{u}_{t-1} + (1 - \beta_1)\hat{g}_t^m$

21:         $v_t^m \leftarrow \beta_2 \bar{v}_{t-1} + (1 - \beta_2)(\hat{g}_t^m)^2$

22:         $\bar{x}_t^m \leftarrow x_t^m - \eta_t \begin{cases} Q_t^m \left[ \dfrac{u_t^m}{\sqrt{v_t^m} + \epsilon} \right] & \text{\color{magenta}No QHM} \\[2em] Q_t^m \left[ \dfrac{\omega u_t^m + (1 - \omega)\hat{g}_t^m}{\sqrt{v_t^m} + \epsilon} \right] & \text{\color{magenta}Low-Rank QHM} \\[2em] (1 - \omega)\dfrac{\hat{G}_t^m}{\mu(\sqrt{v_t^m} + \epsilon)} + \omega Q_t^m \left[ \dfrac{u_t^m}{\sqrt{v_t^m} + \epsilon} \right] & \text{\color{magenta}Full-Rank QHM} \end{cases}$

23:         $\bar{u}_t \leftarrow$ **if** $(t \bmod K_u = 0)$ **then** $\mathbb{E}_m[u_t^m]$ **else** $u_t^m$        Sync $u^j$ every $K_j$

24:         $\bar{v}_t \leftarrow$ **if** $(t \bmod K_v = 0)$ **then** $\mathbb{E}_m[v_t^m]$ **else** $v_t^m$        Sync $v$ every $K_v$

25:         $\Delta_t^m \leftarrow \bar{x}_t^m - x_{t-K_x}^m; \Delta_t \leftarrow \mathbb{E}_m[\Delta_t^m]$        Compute per-worker and aggregated pseudo-gradient

26:         $x_{t+1}^m \leftarrow$ **if** $(t \bmod K_x = 0)$ **then** `OuterOpt`$(\Delta_t, x_{t-K_x}^m)$ **else** $\bar{x}_t^m$        Sync $x$ every $K_x$

# C. Experiment Details

This section details the experimental framework, covering: model architecture and parametrization (Section C.1); the hyperparameter sweep procedure (Section F.3); and the tuning results for LoRDO.

## C.1. Architecture Details and Parametrization

*Table 4.* Model architecture and training hyperparameters used across experiments. We specify the number of (Blocks), attention heads (Heads), embedding dimension ($d_{\mathrm{model}}$), vocabulary size ($|\mathcal{V}|$), and feedforward expansion ratio (Exp. Ratio) for each of the model sizes used in our experiments. Additionally, we show the global batch sizes ($|\mathcal{B}_{\mathrm{G}}|$) and the total number of training steps ($T$) used in our experiments. Our models make use of RoPE positional embeddings (Su et al., 2024), SiLU as the activation function. Additionally, we adopt norm-based gradient clipping with a bound of $\rho$, which are initialized with a typical $\sigma = 0.02$, following guidance from Semenov et al. (2025); Dey et al. (2025). For our optimizers, we use the $\rho$ values recommended by Semenov et al. (2025) for the relevant model scale. Additionally, we set a sequence length that is standard for models at these scales.

| Model Size | Blocks | $d_{\mathrm{model}}$ | $|\mathcal{V}|$ | #Heads | Exp.~Ratio | RoPE $\theta$ | ACT | Init $\sigma$ | $\rho$ | Seq Len | $|\mathcal{B}_{\mathrm{G}}|$ | T |
|---|---|---|---|---|---|---|---|---|---|---|---|---|
| 16M | 4 | 256 | 50K | 4 | 4 | 10000 | SiLU | 0.02 | 1.0 | 2048 | 64 | 4608 |
| 125M | 12 | 768 | 50K | 12 | 4 | 10000 | SiLU | 0.02 | 0.5 | 2048 | 256 | 4608 |
| 720M | 12 | 2048 | 50K | 16 | 4 | 10000 | SiLU | 0.02 | 0.1 | 2048 | 512 | 10240 |

Table 4 summarizes the architectural specifications, which follow standard conventions for large language models of these sizes. To improve training stability and performance, we implement two key architectural modifications. First, we adopt the peri-LayerNorm design (Kim et al., 2025) instead of the standard pre-norm formulation. Second, we use the CompleteP parametrization (Dey et al., 2025) with $\alpha = 1.0$. This configuration enables one-shot hyperparameter transfer from small to large models, allowing us to conduct extensive sweeps on the 16M variant and reserve compute-intensive scaling experiments for baseline comparisons. In Section F, we provide an explanation into how we performed this sweeping procedure to determine the optimal learning rates and coefficients for the quasi-hyperbolic momentum terms.

Batch sizes and training durations follow contemporary best practices (Zhang et al., 2025). For the 125M and 720M models, we adopt global batch sizes of $256$ and $512$, respectively, following Semenov et al. (2025); for the 16M model, we use a batch size of $64$. Training steps $T$ are set as multiples of the compute-optimal token budget (Hoffmann et al., 2022): the 16M model is trained for $\approx 2\times$ the compute-optimal duration, while the 125M and 720M models are trained for $\approx 2\times$ and $\approx 1\times$ their respective budgets. All models utilize the Warmup-Stable-Decay (WSD) scheduler (Hägele et al., 2024). Warmup lengths follow recommendations from Zhang et al. (2025); Hägele et al. (2024); Allal et al. (2025); Semenov et al. (2025), with a fixed warmup period of $T_{\mathrm{WARM}} = 2048$ steps across all experiments.

# D. Additional Experiments

## D.1. Importance of Error Feedback

As discussed in Section 3, in addition to our novel algorithmic improvements, we leverage error-feedback mechanisms (Seide et al., 2014) during the local optimization procedures for each worker to account for information lost during low-rank compression.

In Figure 7, we conduct an ablation on using error feedback across ranks and method types for our 16M model scales. Across all settings, we find error feedback is essential to ensuring good performance, consistent with Seide et al. (2014); Robert et al. (2025).

## D.2. Importance of Well-Positioned Momenta

As mentioned in Section 3, we leverage the insight from Robert et al. (2025), who show that optimizer state rotation is essential once a new projection matrix is determined. We illustrate an ablation across this momenta rotation in our method in Figure 8 for our 16M models for both LoRDO-Global and LoRDO-Local with quasi-hyperbolic momentum terms applied. For both local and global methods, we find that rotating the momenta is essential for performance, corroborating the findings of Robert et al. (2025).

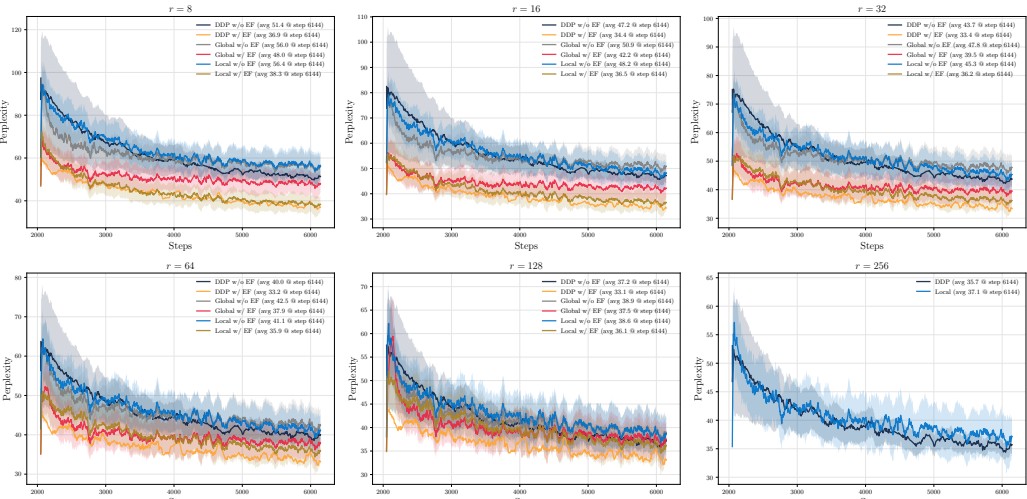

*Figure 7.* Impact of error feedback used during local optimization for `DDP` and the *global* and *local* variants of `LoRDO` across $r \in \{8, 16, 32, 64, 128, 256\}$, where $r = 256$ is the full-rank `Adam` baseline. Across all ranks and model classes, error feedback is essential for optimal performance, corroborating the findings of Seide et al. (2014); Robert et al. (2025).

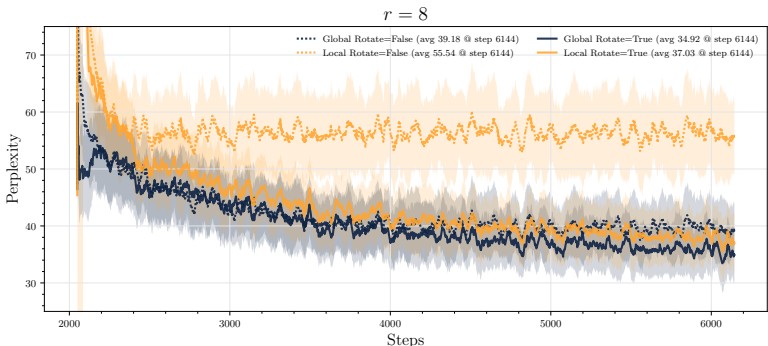

*Figure 8.* Importance of ensuring well-positioned momenta. Both local and global variants of `LoRDO` fail to learn effectively when optimizer states are not rotated into the new basis.

### D.3. Importance of Optimizer State Communication

As mentioned in Section A.7, while communicating the optimizer states in `LoRDO` reduces communication efficiency relative to methods that do not, we show that this is a necessary component for improved performance and robustness, corroborating prior work (Iacob et al., 2026b). As seen in Figure 9, variants of both `LoRDO-Global` and `LoRDO-Local` that do not communicate optimizer states suffer a reduction in performance compared to their standard counterparts. As such, we note that communicating optimizer states presents a trade-off: higher overhead in exchange for stability, and improved performance. `LoRDO` allows practitioners to toggle optimizer state aggregation based on system constraints.

### D.4. Impact of Outer Optimizer on **LoRDO**

In Section E, we note that `LoRDO`, along with similar distributed training methods, falls into the broader `FedOpt` framework (Reddi et al., 2021), which allows for flexible combinations of local and global optimization procedures. In particular, we view `LoRDO` as an innovation targeting the *local optimization procedure*; we use averaging as the base case, as we consider the *global* optimization procedure to be complementary. In Figures 10 and 11, we present initial experiments on integrating alternative choices for the outer optimizer. While Nesterov momentum improves performance when using `Adam` locally, as observed in prior work (Douillard et al., 2023; Charles et al., 2025), this is not the case for `LoRDO`. Specifically, the optimal outer optimizer hyperparameters from Douillard et al. (2023); Charles et al. (2025) do not transfer to the low-rank setting. Furthermore, we note the following:

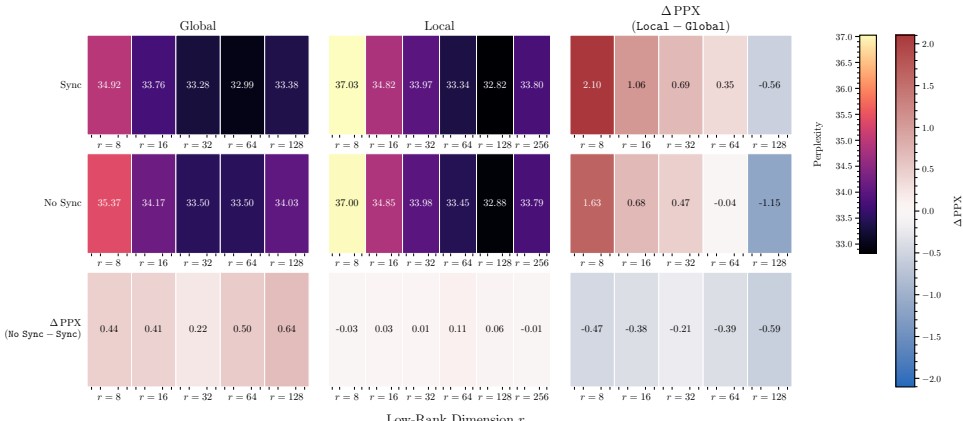

*Figure 9.* Comparison of whether optimizer state synchronization provides improvement for `LoRDO-Global` and `LoRDO-Local` for 16M models. We observe that optimizer states are more important for `LoRDO-Global` relative to `LoRDO-Local`. However, irrespective of whether optimizer states are synchronized, `LoRDO-Global` provides superior performance, particularly at lower ranks.

  i  as with the local optimization procedure, low-rank methods require rotating the outer momenta.
  ii  there is a clear interaction between the hyperparameters chosen for the inner and outer momentum terms.

We leave a detailed investigation of this interaction for future work.

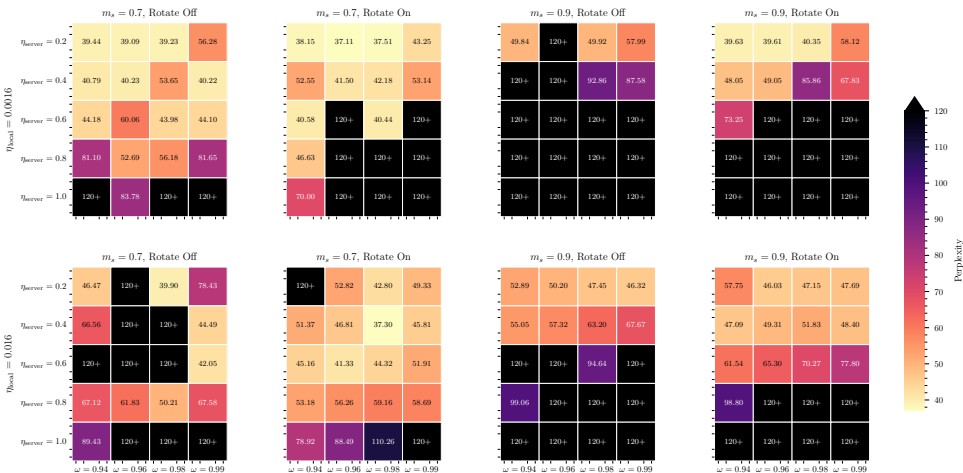

*Figure 10.* Sweeps across server learning rate $\{0.2, 0.4, 0.6, 0.8, 1.0\}$, $\omega \in \{0.94, 0.96, 0.98, 0.99\}$ and the rotation of the sever momentum for `LoRDO-Global` at 16M scale. $\beta_1 = \beta_2 = 0.99$ and $r = 8$. We observe that the optimal server momentum of 0.9 for `DiLoCo` does not transfer to the low-rank optimizer setting. Secondly, we find that rotating the outer momentum is beneficial for `LoRDO`. However, rigourous treatment of the above interaction is left for future work.

### D.5. Comparisons with `DiLoCo`

Supplementing our experiments on the outer optimizer, we provide a direct comparison to `DiLoCo` ([Douillard et al., 2023](#)) at the 16M and 125M scales. To create an effective `DiLoCo` baseline, we made two adjustments:

  i  disabling momentum synchronization ($K_u = K_v \to \infty$).
  ii  setting the outer optimizer to Nesterov.

To determine the optimal outer learning rate, we follow [Charles et al. (2025)](#) by sweeping over $\{0.2, 0.4, 0.6, 0.8, 1.0\}$ and setting the server momentum to 0.9. As shown in [Figure 12](#), `LoRDO` outperforms `DiLoCo` while maintaining a lower local memory overhead (`DiLoCo` requires full-rank `Adam`). Furthermore, our standard `LoRDO` setup (averaging) eliminates the

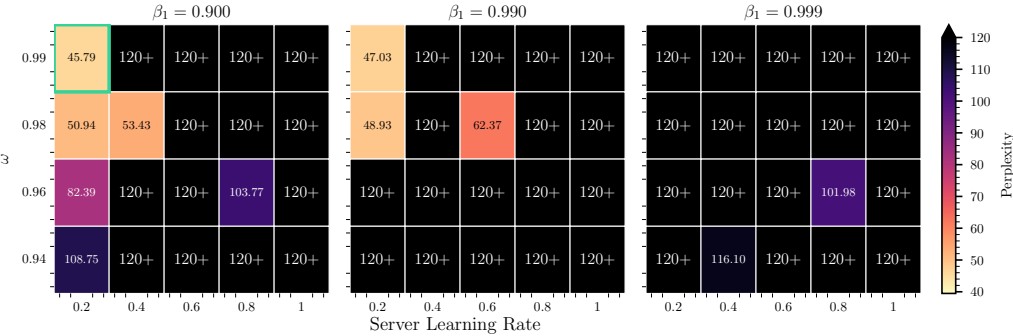

*Figure 11.* Ablation study on the interaction between local optimizer $\beta_1$ values and server momentum. We apply `LoRDO-Global` to a 16M-parameter model with rank $r = 8$. We fix $\beta_2 = 0.999$ and use Nesterov momentum of 0.9 on the server. The ablation evaluates different local momentum parameters $\beta_1 \in \{0.9, 0.99, 0.999\}$ across server learning rates of $\{0.2, 0.4, 0.6, 0.8, 1.0\}$. We observe that lower values of $\beta_1$ interact more robustly with the specific choice of server momentum. This highlights that the server optimizer is complementary to the local optimizer; recommended hyperparameters from full-rank optimizers cannot be naively applied to the low-rank setting.

need to store `DiLoCo`'s additional server momentum term. We posit that this advantage stems from the dual benefit of the quasi-hyperbolic momentum term, corroborating prior work (Iacob et al., 2026a). Specifically, the quasi-hyperbolic momentum term allows the momentum half-lives to match their synchronization intervals, which is not the case in `DiLoCo`. Furthermore, consistent with our findings in Section A.7 and prior work (Iacob et al., 2026b;a), we observe that disabling optimizer state communication harms `DiLoCo`'s performance.

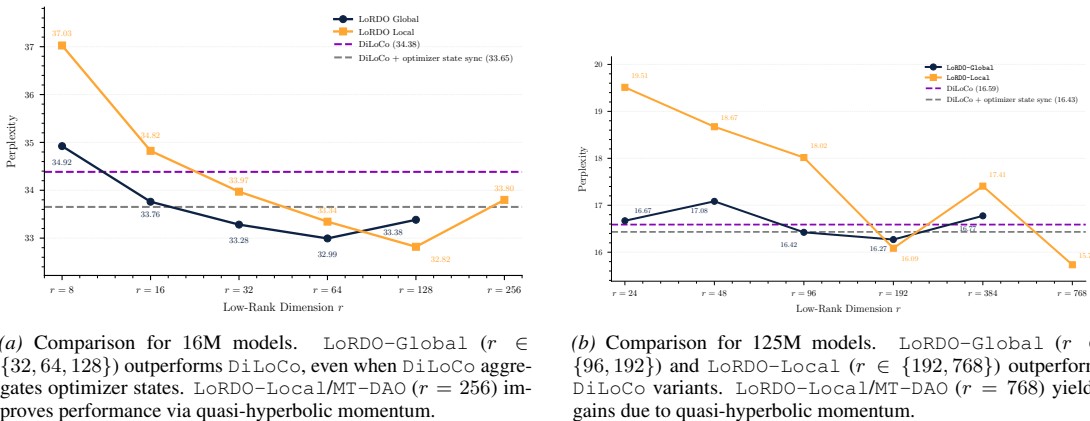

*(a)* Comparison for 16M models. `LoRDO-Global` ($r \in \{32, 64, 128\}$) outperforms `DiLoCo`, even when `DiLoCo` aggregates optimizer states. `LoRDO-Local`/`MT-DAO` ($r = 256$) improves performance via quasi-hyperbolic momentum.

*(b)* Comparison for 125M models. `LoRDO-Global` ($r \in \{96, 192\}$) and `LoRDO-Local` ($r \in \{192, 768\}$) outperform `DiLoCo` variants. `LoRDO-Local`/`MT-DAO` ($r = 768$) yields gains due to quasi-hyperbolic momentum.

*Figure 12.* Performance comparison between `LoRDO` and `DiLoCo` across 16M and 125M model scales.

### D.6. Matched Communication Budget

The primary mechanism by which `LoRDO` reduces communication is the compression of transmitted optimizer states. However, a similar reduction could be achieved by constraining the communication budget of a full-rank optimizer to match that of `LoRDO`, independent of the memory overhead advantages of `LoRDO`. To evaluate this, we adjust the synchronization intervals $(K_x, K_u, K_v)$ of the full-rank baseline to match the communication footprints of both `LoRDO-Global` and `LoRDO-Local`. Specifically, we examine two configurations:

  i a uniform setting where we fix $K_x = K_u = K_v$, and
  ii a skewed setting where we fix $K_x = 32$ and select $K_u, K_v$ accordingly to match the budget.

These synchronization intervals chosen such that the equate the communication overhead accounted for the communication savings introduced by `LoRDO`.

As illustrated in Figure 13, the uniform (*fixed*) synchronization strategy inflates the perplexity of the full-rank method

compared to its baseline. Consequently, LoRDO outperforms this baseline across all ranks except $r = 8$. This performance degradation in the communication-matched baselines is due to the fact that model parameters and momentum terms are communicated too infrequently. We note that the perplexity increase is less pronounced at larger values of $r$, where the payload gap between the full-rank method and LoRDO is relatively smaller, allowing the full-rank method to afford more frequent synchronization. In line with prior work (Iacob et al., 2026b), we observe that allowing for more frequent communication of the model parameters ($K_x = 32$) makes the performance hit less pronounced. However, in this case, even when communication budgets are perfectly matched, LoRDO still provides an up to an $8\times$ reduction in optimizer state memory overhead under the relevant settings of $r$.

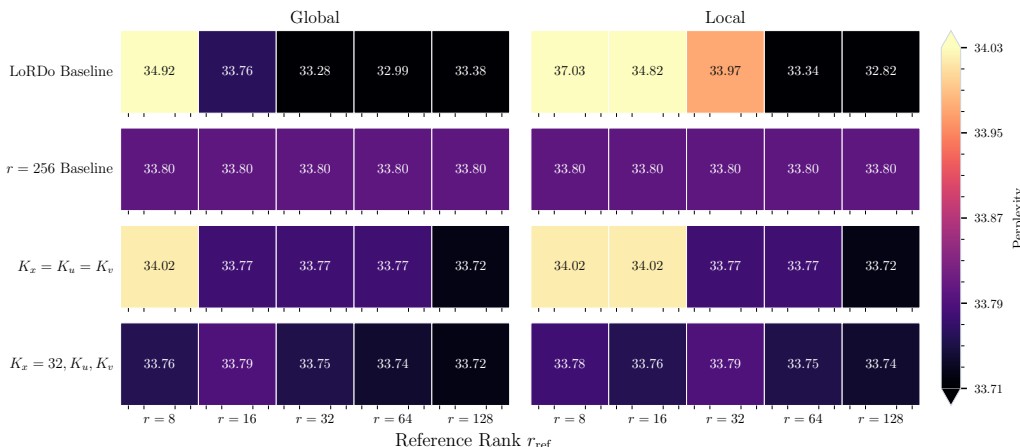

*Figure 13.* Comparison of LoRDO-Global and LoRDO-Local against a full-rank communication-matched baseline for 16M models. Communication is matched either by fixing $K_x = K_u = K_v$, or by setting $K_x = 32, K_u, K_v$ of the $r = 256$ baseline, where the values are determined to match the overhead of LoRDO-Global (right) and LoRDO-Local (left). LoRDO-Global is superior for all ranks except $r = 8$, where the communication-matched full-rank model suffers a degradation in performance as the model parameters and momenta terms are communicated too infrequently.

### D.7. LoRDO Impact on Wall-Clock Time

Based on the theoretical framework presented in Section A.5, the communication efficiency of low-rank optimizers is expected to provide improved wall-clock performance relative to DDP baselines. To show this empirically, we report the wall-clock time of our 720M parameter experiments under both the high bandwidth of our experimental setup (Figure 14a) and a theoretically modeled lower-bandwidth environment (Figure 14b). This is done to highlight that the advantages of communication-efficient distributed training become increasingly pronounced as the bandwidth between the workers decreases.

As illustrated in (Figure 14), we first observe that, within the same training paradigm (standard DDP or communication-efficient), low-rank optimizers introduce a wall-clock time overhead despite their lower memory footprint. As noted by Robert et al. (2025), this is due to the additional computation cost associated with the low-rank projection operation. Second, transitioning from DDP to a communication-efficient regime consistently improves wall-clock time for both full-rank and low-rank optimization by minimizing synchronization frequency ($K = 1 \rightarrow K = 32$ in this setting). Finally, these performance gains are more pronounced in lower-bandwidth regimes (Figure 14b) as expected, where communication-efficient methods like MT-DAO and LoRDO significantly outperform DDP.

### D.8. DDP Gradient versus LoRDO-Global

In Section 5, we observe that, particularly at lower ranks, LoRDO-Global significantly outperforms its DDP variant. We posit two factors causing these differences:

  i Allowing a larger exploration of the loss landscape (i.e., having a greater history due to the $K$ step window) provides a more informative signal than the gradient.

  ii Integrating curvature information into the projection matrix computation creates more informative bases. Mathemati-

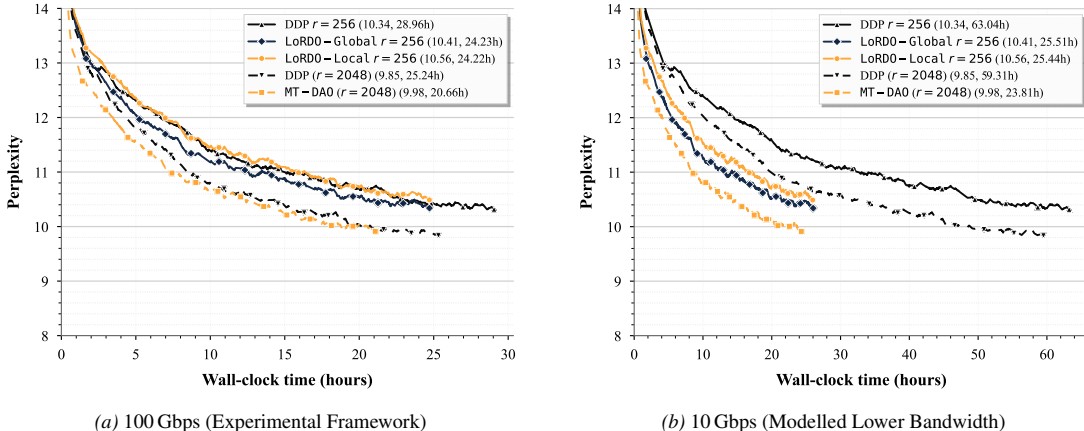

*(a)* 100 Gbps (Experimental Framework)  *(b)* 10 Gbps (Modelled Lower Bandwidth)

*Figure 14.* Perplexity versus wall-clock time for LoRDO against baselines at the 720M scale across high and low bandwidth settings. Low-rank methods trade wall-clock overhead for lower memory usage. Communication-efficient training improves wall-clock time for both full- and low-rank methods, with MT-DAO and LoRDO showing the largest gains over DDP in low-bandwidth regimes.

cally, the SVD on raw gradients ($g \approx -Hp^*$) gets dominated by the steep, high-curvature directions of $H$. Applying second-order information ($-H^{-1}g \approx p^*$) cancels out this curvature distortion, forcing the SVD to project directly onto the subspace of true optimal steps.

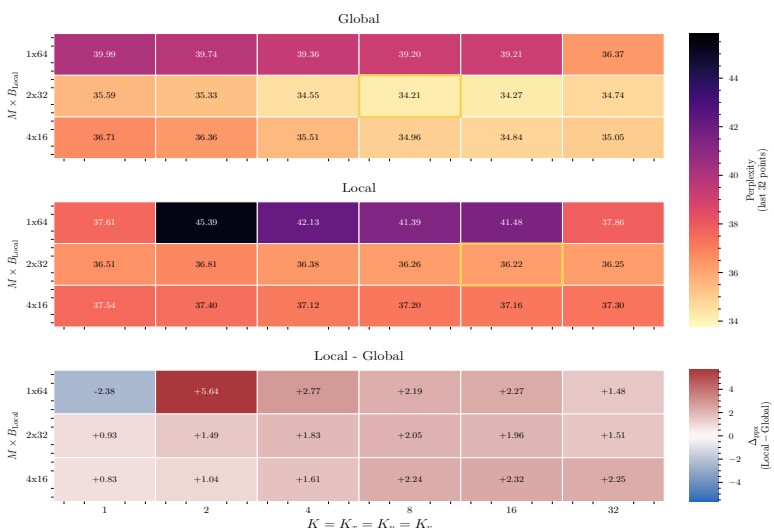

*Figure 15.* Comparison of LoRDO-Global and LoRDO-Local for 16M models where $r = 8$ across the $M \times B_{\text{Local}}$ axis for $B_{\text{Global}} = 64$. Additionally, we fix $K = K_x = K_u = K_v$ for simplicity. We consider $M \in \{1, 2, 4\}$ to avoid concerns of small local batch sizes as per Figure 4. In all regimes where $K > 1$, i.e. the regimes of interest for communication-efficient training, LoRDO-Global provides superior performance over LoRDO-Local.

In Figure 15, we visualize both axes of comparison. Isolating the impact of the local steps first: for a fixed global batch size, we evaluate both LoRDO-Global and LoRDO-Local across synchronization intervals $K \in \{1, 2, 4, 8, 16, 32\}$. In the case of LoRDO-Local, at a fixed batch size, we find effectively no difference when increasing the frequency at which subspaces are computed; the gradient signal remains consistently informative along this axis. However, with LoRDO-Global, performance improves as synchronization is delayed. We posit that this process allows each worker to explore a larger region of its loss landscape. Consequently, upon aggregation, this creates a signal that is more informative for the projection computation, as it possesses a greater history of information for determining the optimal basis.

For the second component, we view Figure 15 but focusing on the single-worker case across synchronization frequencies. Although LoRDO-Global uses a single worker to determine its projection matrix (after 32 steps of local training), its

has an improved perplexity relative to its `DDP` counterpart (`LoRDO-Local` in a single worker regime). This suggests that the curvature information baked into the pseudo-gradients of each worker helps determine a more informative projection matrix, especially considering `LoRDO`'s performance improvement as the number of workers increases. This corroborates similar findings in Figures 2a and 4. We leave a thorough investigation on the interplay between projecting gradients and pseudo-gradients to future work.

### D.9. Sparse Communication Impact on `LoRDO`

Although methods implementing communication compression at synchronization intervals are orthogonal to our work, we consider the effect of sparsification to additionally improve the communication overhead for `LoRDO`. In Figure 16, we ablate across percentages of `Top-K` sparsity for both the global and local variants. As expected, we find that with increased sparsification, the performance of `LoRDO` degrades regardless of whether projection matrices are determined locally or globally. There is less meaningful signal for the outer optimizer to learn a representation beneficial for the global optimization trajectory. Interestingly, we find that `LoRDO-Global` is not more affected by sparsity than its local counterpart. We posit two reasons for this behavior. First, even though individual pseudo-gradients are sparse, their aggregation creates an informative signal—not necessarily sparse (Guastella et al., 2025)—that preserves the dominant directions necessary to determine a robust global projection matrix. Second, unlike the local variant, the `LoRDO-Global` method employs full-rank quasi-hyperbolic momentum terms. This injects a full-rank signal into the update, allowing the optimizer to correct for meaningful directions potentially lost during the sparsification of the projection subspace. We leave a full investigation into the combination of communication compression and `LoRDO` for future work.

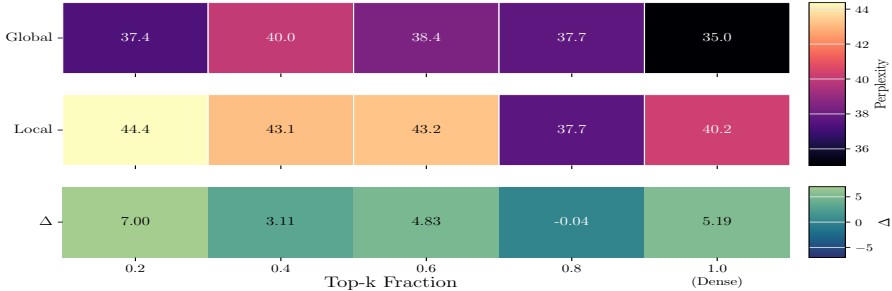

*Figure 16.* Ablation across sparsity levels for `LoRDO-Global` and Local, reporting the final perplexity for each, along with the difference $\Delta = PPX_{\text{Local}} - PPX_{\text{Global}}$. We observe that while both methods are affected by increasing sparsity, the addition of the full-rank quasi-hyperbolic momentum signal allows for improved performance relative to `LoRDO-Local`, despite using a potentially damaged projection matrix due to pseudo-gradient sparsification.

### D.10. Full 720M Results

In Section 5, we present the results for the 720M scale at the end of training for brevity. In Figure 17, we present the full set of results across the entire duration of training:

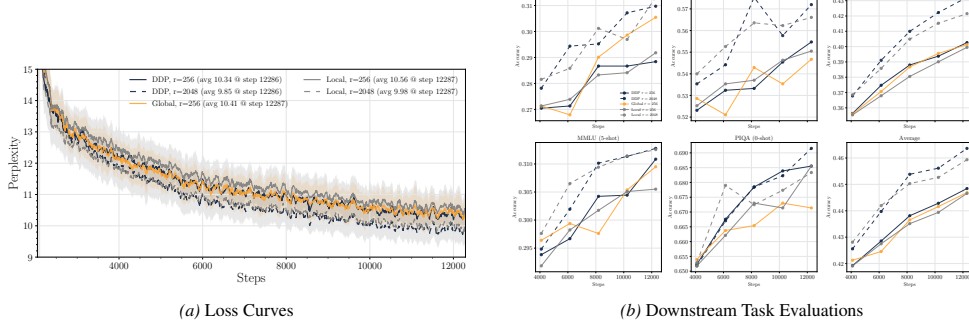

*(a)* Loss Curves          *(b)* Downstream Task Evaluations

*Figure 17.* Comparison of `LoRDO` and `DDP` at the 720M scale. Figure 17a shows that `LoRDO` tracks its `DDP` counterpart consistently, a trend reflected across downstream tasks. While a performance gap between the low-rank and full-rank optimizers remains, we posit that with extended training, `LoRDO` converges toward full-rank `DDP` performance.

# E. Extended Related Work

**Memory-Efficient Optimization and Communication Compression.**  Significant research targets communication volume reduction by compressing the synchronization payload rather than altering the local optimization state representation. Techniques such as quantization (Alistarh et al., 2017) and sparsification (Lin et al., 2018) reduce the bits transmitted per synchronization. In LLM training, `CocktailSGD` (Wang et al., 2023) combines random sparsification, top-$k$ selection, and quantization. `DeMo` (Peng et al., 2024) employs error feedback with top-$k$ compression on gradients. In the federated setting, `Streaming DiLoCo` (Douillard et al., 2025) and `MuLoCo` (Thérien et al., 2026) apply compression to the pseudo-gradients exchanged between workers. `DiLoCoX` (Qi et al., 2025) integrates pipeline parallelism with a dual optimizer policy, employing low-rank compression on pseudo-gradients after local training. Similarly, `SparseLoCo` (Sarfi et al., 2025) replaces `DiLoCo`'s global outer momentum with a local error-feedback accumulator to enable aggressive top-$k$ sparsification. We leave the integration of these compression techniques with our low-rank framework for future work.

**Communication Mechanics of Low-Rank Updates.**  Beyond standard compression, distinct communication trade-offs arise depending on the specific low-rank strategy employed. In the *local* method, utilizing either low-rank quasi-hyperbolic updates or standard updates provides a direct communication benefit, as only the low-rank matrices of the pseudo-gradient need to be transmitted. For the *global* method without quasi-hyperbolic updates, the communication benefit increases as transmitting projection matrices becomes unnecessary; however, this approach is limited to a fixed rank-$r$ subspace, potentially causing learning stagnation. Conversely, the *global* method with quasi-hyperbolic momentum requires transmitting the full pseudo-gradient to enable the server to form a new basis. While this forfeits the intrinsic low-rank communication reduction for the gradient itself, it ensures robust subspace exploration. Crucially, this full-rank transmission remains fully compatible with other compression methods from prior work, such as quantization and sparsification, and we leave the study of this composition for future work.

**Alternative Subspace Estimation Techniques.**  Beyond SVD (used in `GaLore`) and Block Power Iteration (used in `LDAdam`), other methods exist for estimating significant gradient subspaces. FFT-based Subspace Selection (Modoranu et al., 2025) offers a computationally cheaper alternative using the Discrete Cosine Transform (DCT) matrix to select columns dynamically based on alignment with the gradient. `Dion` (Ahn et al., 2025) also falls into this category by enabling rank-$r$ orthogonalization; however, it typically relies on QR decomposition, which requires execution at every step, incurring significant computational costs that scale with rank (Modoranu et al., 2025). While we primarily discuss SVD-based projections, our framework's core contribution—handling subspace misalignment in infrequent synchronization—is agnostic to the specific estimation method. We leave the exploration of integrating these faster or alternative subspace estimators into our distributed framework for future work.

**Connection to the FedOpt Framework.**  Our approach fits within the `FedOpt` (Reddi et al., 2021) abstraction, which generalizes Federated Averaging (`FedAvg`) (McMahan et al., 2017) by allowing the server (outer optimizer) to maintain its own state. `Mime` (Karimireddy et al., 2020) represents another point in this design space, using control variates to reduce client drift. Our innovation specifically modifies the *internal structure* of the client-side optimizer states (projecting them to low-rank) rather than just the aggregation logic or control variates, offering a new dimension for optimization efficiency within the FedOpt paradigm. Crucially, the underlying principle of projecting local optimizer states into a low-rank subspace is optimizer-agnostic; while we focus on adaptive methods, the framework generalizes to other stateful optimizers. The investigation of other outer optimizers or control variates within our low-rank framework is left for future work.

# F. Hyperparameter Tuning and Warm-up Procedure

Below, we detail the hyperparameter tuning procedure for both `DDP` and `LoRDO`. For all the non-quasi-hyperbolic experiments, we follow previous literature that establishes that learning rates transfer effectively between `DDP` and the distributed setting (Iacob et al., 2026b). This allows us to ensure that all models achieve the best performance possible in the `DDP` setting, which then transfers to `LoRDO` in the non-quasi-hyperbolic case.

Given that the quasi-hyperbolic formulation is only activated following the warm-up period, although `DDP` and `LoRDO` follow the same tuning methodology, we conduct independent tuning sweeps to ensure that the comparisons across `DDP` and `LoRDO` were fair and consistent, given the differences in the two methods. The procedure that we implement for both methods follows the two-phase approach as per Iacob et al. (2026a). Specifically:

i **Choosing best warm-up hyperparameters.** Both `DDP` and `LoRDO` adopt the best hyperparameters for the optimizer for the pre-warm-up phase to ensure that the baselines at the warm-up cut-off are as strong as possible.

ii **Choosing hyperparameters post warm-up.** At the end of the warm-up, both `DDP` and `LoRDO` use the same model state produced by the warm-up checkpoint. Then, each method independently tunes the hyperparameters (specifically, the new learning rate $\eta$ and the $\omega$ coefficient for quasi-hyperbolic momenta) to determine the optimal setting in each. This is effectively done in the stable portion of the `WSD` schedule. The new learning rate is found by searching across multiples of the base learning rate applied with a "switch scale". For both the `DDP` and `LoRDO` tuning, we fix $\beta_1 = 0.999$ following warm-up phase in the case of the quasi-hyperbolic experiments.

In the case of all methods, we sweep over $\omega \in \{0.90, 0.91, 92, 0.93, 0.94, 0.95, 0.96, 0.97, 0.98, 0.99\}$. For the learning rates, we consider grids in increasing powers of two to find the optimal value as we show in Sections F.3 to F.5. Below, we provide a table summarizing the optimal hyperparameters for our experimental setup. Specifically, these hyperparameter values then transfered one-shot to the larger model sizes given the `CompleteP` framework. In the case of low-rank optimizers, this is done for each ratio of $r/d_{\text{model}}$ that we consider in our experiments to ensure that we do not violate the assumptions; please see a larger discussion in Section F.1.

*Table 5.* Best hyperparameters found from sweeps across `DDP` and `LoRDO` variants, where rank is expressed as a ratio of $d_{\text{model}}$.

| $r/d_{\text{model}}$ | $\eta$ | DDP | | | LoRDO-Global | | | LoRDO-Local | | |
|---|---|---|---|---|---|---|---|---|---|---|
| | | $\omega$ | QHM Form | $\eta^* = \eta \times$ Switch Scale | $\omega$ | QHM Form | $\eta^* = \eta \times$ Switch Scale | $\omega$ | QHM Form | $\eta^* = \eta \times$ Switch Scale |
| 1/32 | 0.016 | 0.94 | Low-Rank | 0.032 | 0.99 | Full-Rank | 0.016 | 0.90 | Low-Rank | 0.032 |
| 1/16 | 0.008 | 0.94 | Low-Rank | 0.016 | 0.97 | Full-Rank | 0.004 | 0.92 | Low-Rank | 0.016 |
| 1/8 | 0.008 | 0.91 | Low-Rank | 0.016 | 0.97 | Full-Rank | 0.004 | 0.94 | Low-Rank | 0.016 |
| 1/4 | 0.008 | 0.94 | Low-Rank | 0.008 | 0.97 | Full-Rank | 0.004 | 0.98 | Low-Rank | 0.016 |
| 1/2 | 0.016 | 0.92 | Low-Rank | 0.016 | 0.97 | Full-Rank | 0.008 | 0.93 | Low-Rank | 0.008 |
| 1 | 0.008 | 0.94 | Low-Rank | 0.008 | $--$ | $--$ | $--$ | $--$ | $--$ | $--$ |

### F.1. Adopting `CompleteP`

We adopt the `CompleteP` framework to ensure that our optimal hyperparameters scale across model size, following prior work (Iacob et al., 2026a). Although this extends naturally to distributed training, and the application of low-rank optimizers, below we provide a mathematical argument supporting this claim. As per Yang et al. (2021), effective scaling must ensure that hidden pre-activations are model-scale invariant. Specifically, if $h = Wx$ (where $W \in \mathbb{R}^{d \times d}$ and $x \in \mathbb{R}^d$), the change in pre-activations after an update step $\Delta W$ is:

$$\Delta h = (\Delta W)x \tag{30}$$

`CompleteP`-like frameworks (Dey et al., 2025) ensure that as $d \to \infty$, the magnitude of this update remains perfectly bounded with increasing model size:

$$\Delta h = \Theta(1) \tag{31}$$

#### F.1.1. LOW-RANK OPTIMIZERS

Low-rank optimizers project updates onto a lower-dimensional subspace of rank $r$. In this regime, the magnitude of the low-rank update, relative to a full-rank update, shrinks by an $r$-dependent factor:

$$\Delta W_{LR} = \sqrt{\frac{r}{d}}\Delta W_{full} \tag{32}$$

As such, the change in pre-activations under the low-rank optimizer becomes:

$$\Delta h_{LR} = \left(\sqrt{\frac{r}{d}}\Delta W_{full}\right)x = \sqrt{\frac{r}{d}}(\Delta h_{full}) \tag{33}$$

This could potentially violate the `CompleteP` requirement of $\Delta h = \Theta(1)$ if $r$ were frozen as $d$ increased with model scale. However, we tuned the learning rates using a fixed width-to-rank ratio for the low-rank optimizer versions across all model scales. We define this as a constant $\rho$:

$$\frac{d}{r} = \rho \implies \sqrt{\frac{r}{d}} = \frac{1}{\sqrt{\rho}} \tag{34}$$

Applying this constant into the pre-activation update equation yields:

$$\Delta h_{LR} = \frac{1}{\sqrt{\rho}}(\Delta h_{full}) \tag{35}$$

Since $1/\sqrt{\rho}$ is a positive, width-independent constant, it follows that:

$$\Delta h_{LR} = \frac{1}{\sqrt{\rho}}\Theta(1) = \Theta(1) \tag{36}$$

This proves that low-rank optimizers preserve the scaling for `CompleteP`. The scalar $1/\sqrt{\rho}$ acts as a global multiplier that is absorbed into the base learning rate.

### F.1.2. COMMUNICATION-EFFICIENT REGIMES

After $K$ steps, the local worker model $m$ can be expressed as the original weights plus the accumulated local updates:

$$W_m = W_0 + \Delta W_m \tag{37}$$

When the $M$ local models are aggregated at the communication step, the new global weights are obtained via averaging:

$$W_{new} = \frac{1}{M} \sum_{m=1}^{M} W_m \tag{38}$$

$$W_{new} = \frac{1}{M} \sum_{m=1}^{M} (W_0 + \Delta W_m) = W_0 + \frac{1}{M} \sum_{m=1}^{M} \Delta W_m \tag{39}$$

From this, the effective global update is $\Delta W_{global} = \frac{1}{M} \sum_{m=1}^{M} \Delta W_m$. To prove that hyperparameter transfer holds, we must show that the global change in pre-activations ($\Delta h_{global}$) remains model-scale invariant. We can express this global feature update as:

$$\Delta h_{global} = (\Delta W_{global})x = \left(\frac{1}{M} \sum_{m=1}^{M} \Delta W_m\right) x = \frac{1}{M} \sum_{m=1}^{M} (\Delta W_m x) = \frac{1}{M} \sum_{m=1}^{M} \Delta h_m \tag{40}$$

Because the number of workers $M$ is a finite constant independent of the model width $d$, and because CompleteP guarantees that every local update satisfies $\Delta h_m = \Theta(1)$, it mathematically follows that:

$$\Delta h_{global} = \frac{1}{M} \sum_{m=1}^{M} \Theta(1) = \Theta(1) \tag{41}$$

This reflects the linear combination primitive in Yang (2021). Averaging is strictly a linear operation over width-independent combinations; no width-dependent scaling factors are introduced or destroyed. If `CompleteP` dictates a specific $1/d$ scaling for a layer's update magnitude to achieve $\Theta(1)$ feature learning, the aggregated update strictly preserves this limit, guaranteeing zero-shot transfer without modification.

## F.2. Findings: Ablation on Low- and Full-Rank QHM

In addition to the hyperparameter sweeps for global and `LoRDO-Local`, Figure 22 provides us with an ablation across both low- and full-rank quasi-hyperbolic momentum updates for two `LoRDO` variants. Focusing on `LoRDO-Global`, we first observe that adding quasi-hyperbolic momentum terms in their low-rank form does alleviate the learning stagnation. The additional gradient signal is still confined to the low-rank subspace induced by the projection matrix. This is unlike the full-rank gradient signal, which allows `LoRDO-Global` to explore the full $d$ dimensional representation space. Furthermore, we find that the difference between the two variants (low- and full-rank QHM `LoRDO-Global`) decreases as the rank increases, as expected; as there is less compression, `LoRDO-Global` can explore more dimensions within its representation.

For `LoRDO-Local`, we observe the opposite: there is very little difference as to whether one applies the quasi-hyperbolic momentum term in its low- or full-rank form. The reason for this is that, irrespective the local optimization signal, the pseudo-gradient signal still recovers its full-rank structure following aggregation across workers. Furthermore, the full-rank variant is, at best, as performant as low-rank QHM for `LoRDO-Local`. As such, we use low-rank QHM for `LoRDO-Local` and full-rank QHM for `LoRDO-Global`, respectively, throughout our experiments.

## F.3. Non-Quasi-Hyperbolic Sweeps

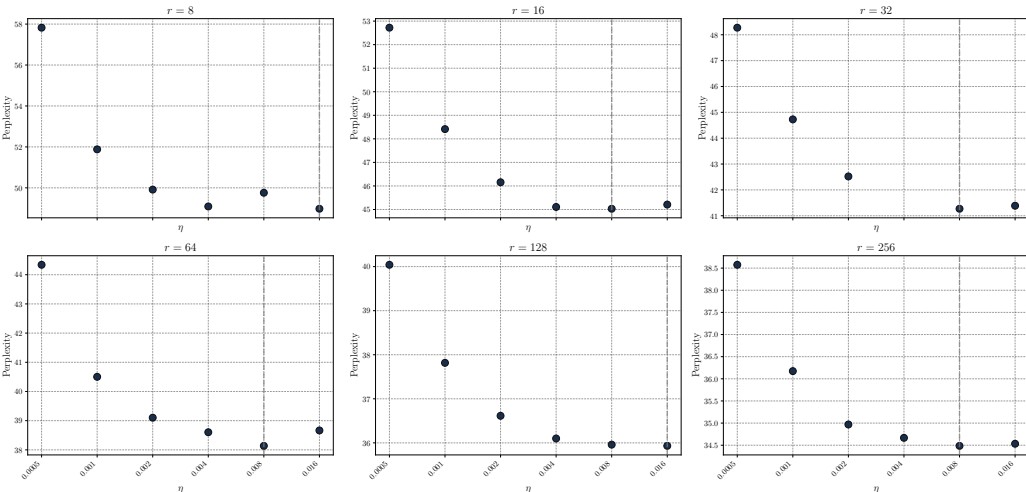

*Figure 18.* Hyperparameter sweep across $\eta \in \{0.0005, 0.001, 0.002, 0.004, 0.008, 0.016\}$ for 16M models to determine optimal learning rate for warmed up model training starting point.

## F.4. DDP Quasi-Hyperbolic Sweeps

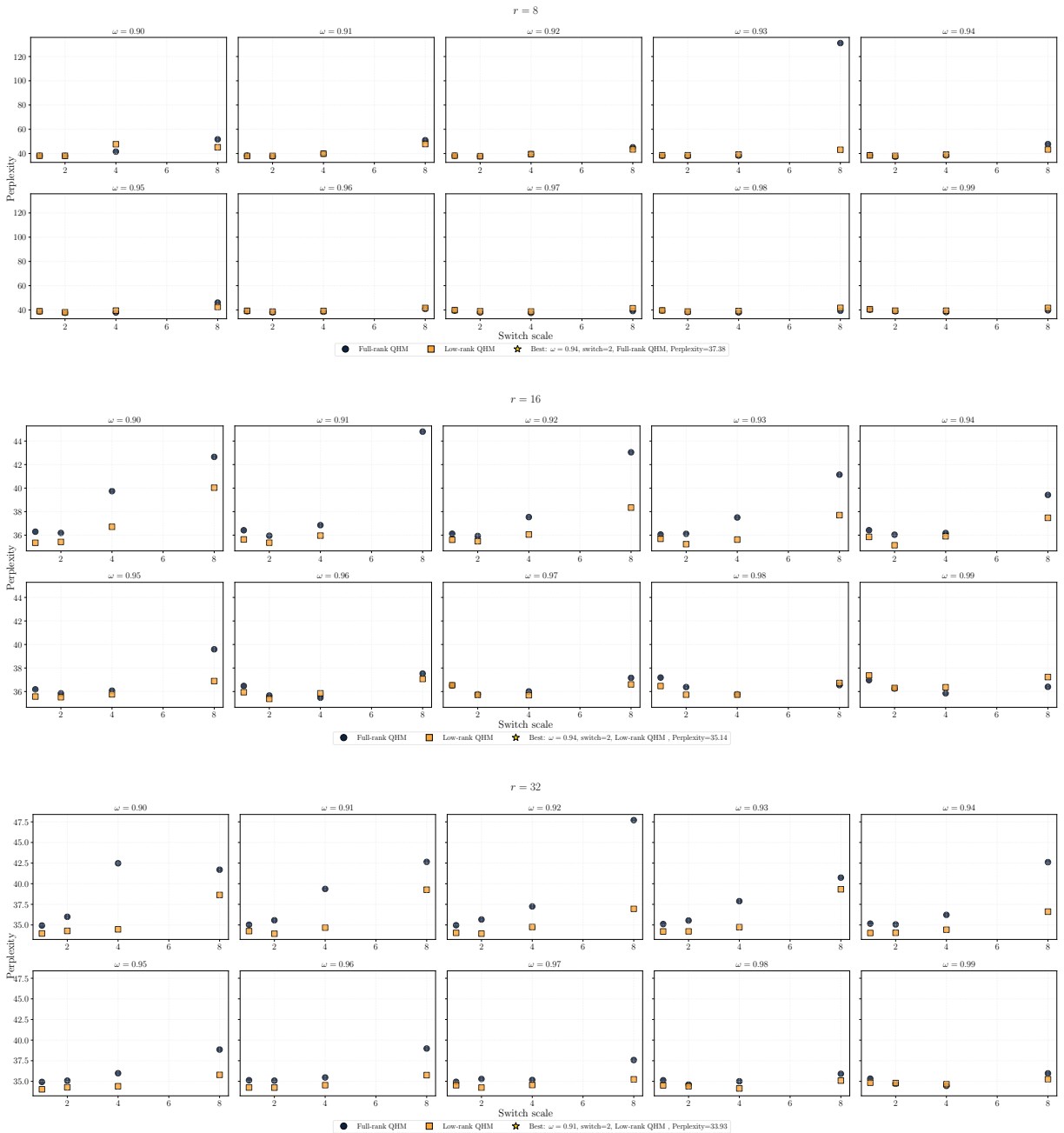

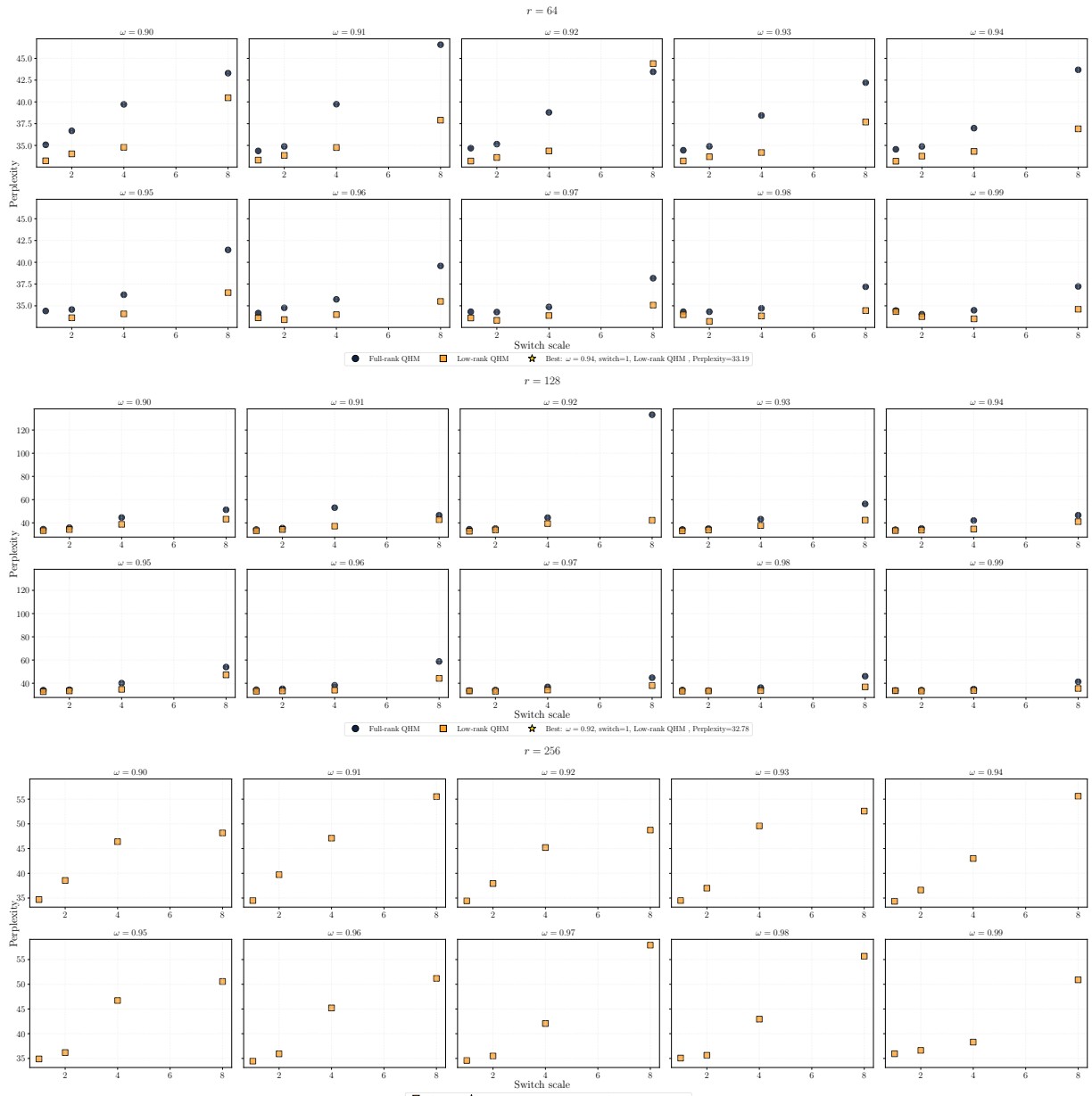

*Figure 20.* Hyperparameter sweep for 16M models trained with DDP, where $\beta_1 = \beta_2 = 0.999$ across $r \in \{8, 16, 32, 54, 128\}$ for combinations of $\omega \in [0.90.99]$ and different multiples (switch scale) of the learning rate as per (Iacob et al., 2026a). For sweep combination, we show the effect of applying the quasi-hyperbolic formulation in its full-rank or low-rank form. Due to the approximation that the full-rank QHM method uses, it has a larger tendency to diverge quicker in regimes where it is not well-tuned, relative to the low-rank counterpart. However, in well-tuned regimes, there is little difference between the full or low-rank QHM versions, although the low-rank QHM is consistently superior. This corroborates earlier findings where our LoRDO's local variant does not benefit from a full-rank QHM formulation, unlike the global counterpart.

## F.5. LoRDO Quasi-Hyperbolic Sweeps

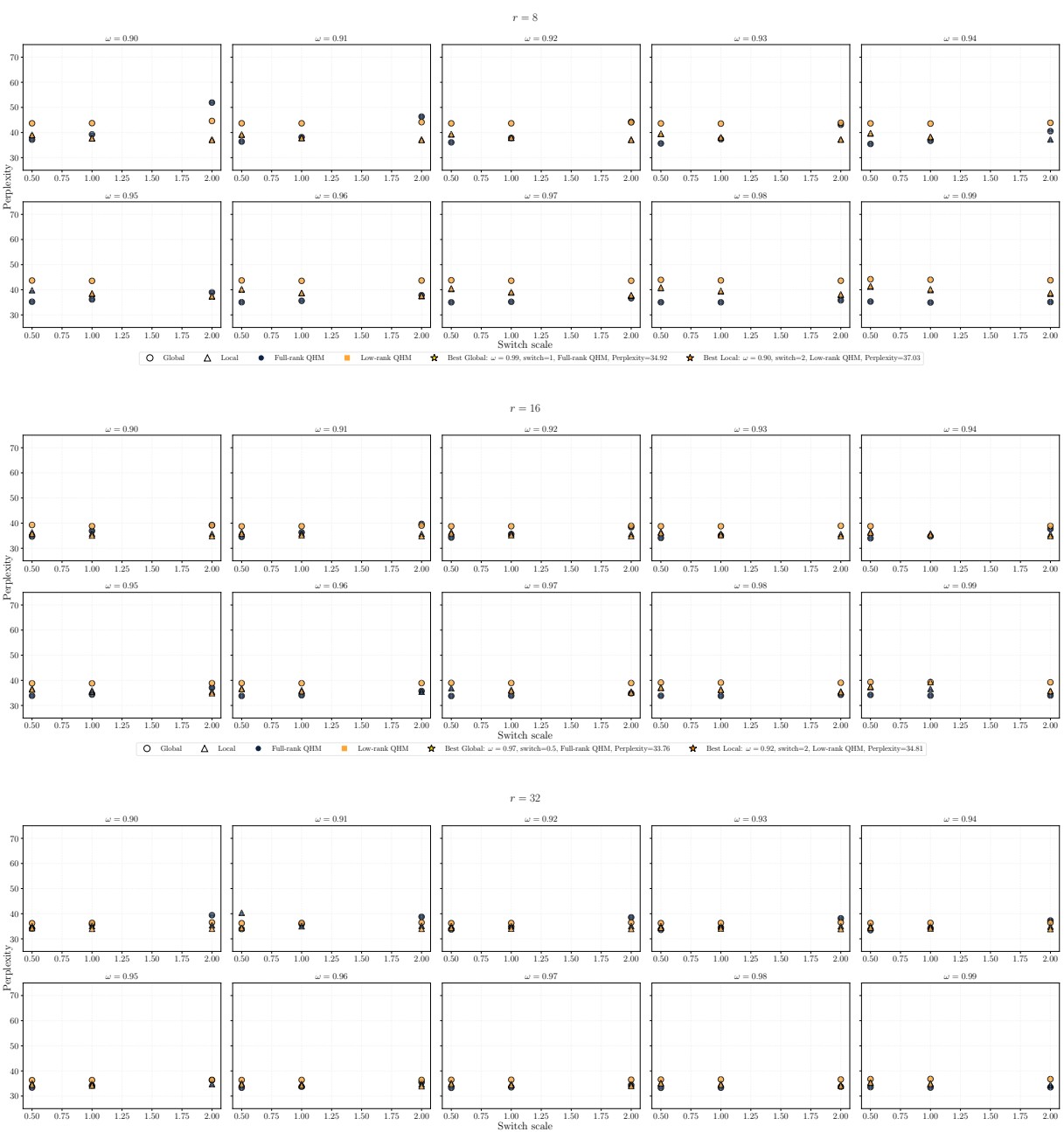

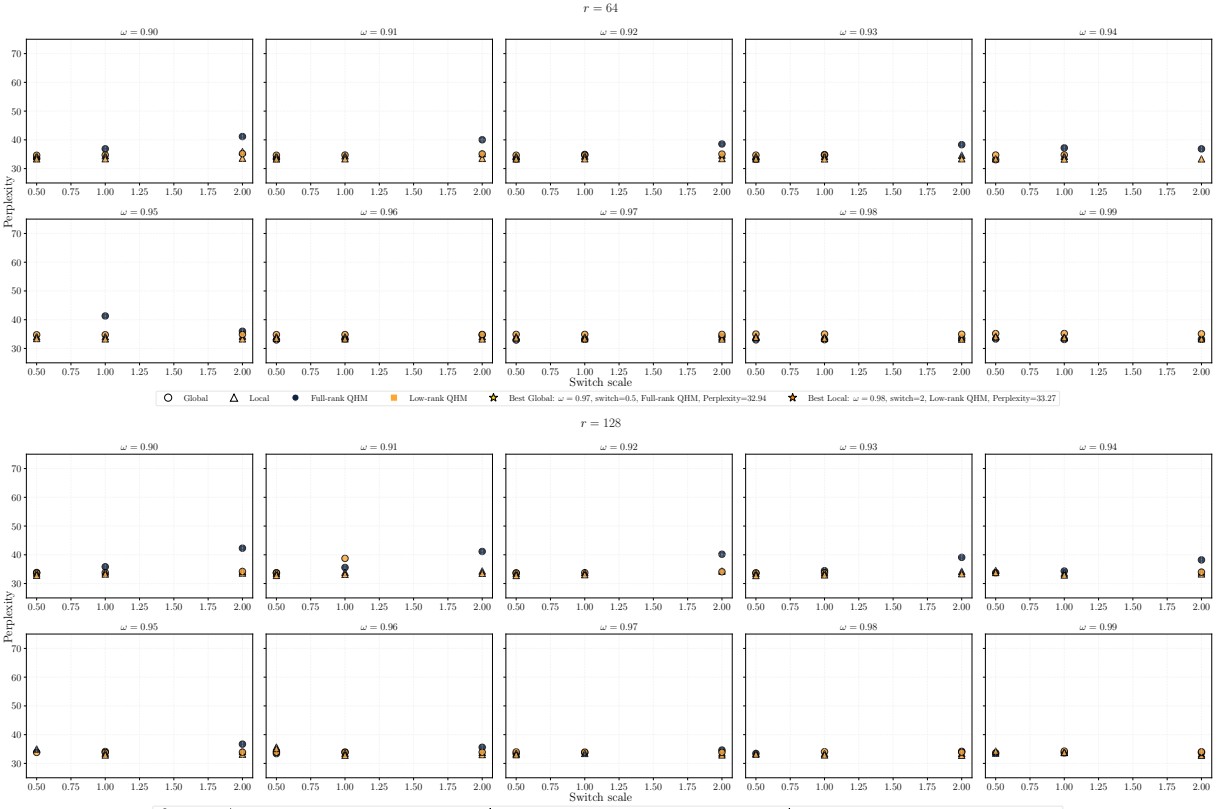

*Figure 22.* Hyperparameter sweep for 16M models trained with LoRDO, where $\beta_1 = \beta_2 = 0.999$ across $r \in \{8, 16, 32, 54, 128\}$ for combinations of $\omega \in [0.90.99]$ and different multiples (switch scale) of the learning rate as per (Iacob et al., 2026a). For sweep combination, we show the effect of applying the quasi-hyperbolic formulation in its full-rank or low-rank form. As in Figure 20, due to the approximation that the full-rank QHM method uses, it has a larger tendency to diverge quicker in regimes where it is not well-tuned, relative to the low-rank counterpart. However, in well-tuned regimes, the full-rank QHM is essential to ensuring good performance for LoRDO-Global, where LoRDO's local variant does not benefit from a full-rank QHM formulation.

