# OpenReview forum: "LoRDO: Distributed Low-Rank Optimization with Infrequent Communication"
_ICML.cc/2026/Conference — ICML 2026 regular_

### Official Review · Reviewer_UTAu · 2026-03-05

**Soundness:** 3
**Presentation:** 2
**Significance:** 2
**Originality:** 3
**Overall Recommendation:** 5
**Confidence:** 3

**Summary:**

LoRDO combines low-rank adaptive optimization with distributed training under infrequent communication. It addresses the authors observation that local low-rank projections are noisy because each worker only sees a small local batch, while global projections from aggregated low-rank gradients are more stable but can trap optimization in a fixed low-rank subspace. Authors address this by combining a global projector with a full-rank "quasi-hyperbolic update" (QHM) injection, which mixes a low-rank momentum-based step with a direct full-rank gradient step so the projection subspace can keep evolving. Experiments are on 16M-720M decoder-only transformers trained on SmolLM2 and show performance close to low-rank DDP while reducing theoretical communication cost and optimizer-state memory.

**Compliance With Llm Reviewing Policy:**

Affirmed.

**Final Justification:**

While some questions about practical adoption relative to simpler alternatives remain, the rebuttal addressed my main concerns with additional experiments and clarifications, particularly on wall-clock behavior, matched communication budgets, and the role of QHM versus the projector. Given these improvements, I am raising my score to 5.

**Key Questions For Authors:**

1. Can the authors provide wall-clock or throughput comparisons? Since the main motivation is reduced communication in distributed training, it would be important to show whether the communication savings translate into faster training or improved time-to-target.
2. Can the authors compare against a matched communication-budget baseline where synchronization is simply performed less frequently? Instead of reducing payload via low-rank communication, what happens if one increases the synchronization interval K proportionally? This would help clarify when low-rank communication is preferable to a simpler reduced-sync strategy.
3. How much of the gain comes from the LoRDO projector mechanism itself versus the more general effect of mixing momentum with fresh gradient information? It would strengthen the paper to test similar gradient-injection mixtures as baselines for both low-rank and full-rank optimizers.

**Limitations:**

Authors list concrete limitations in Appendix A that include: limited 720M validation across ranks, no empirical confirmation of the derived communication/memory bounds, and lack of experiments validating the Non-IID interference hypothesis.
Since this part is important, I would suggest moving a short version of these limitations into the main paper

**Strengths And Weaknesses:**

Soundness/originality: The paper appears technically solid and combines existing approaches in a meaningful way. The failure mode of global projections without QHM is clearly explained and supported empirically. The experimental framework is also reasonably comprehensive, with explicit research questions and analyses of rank, batch size, and synchronization effects.

My main weaknesses are in significance and presentation. The paper emphasizes reduced communication, but the experiments do not convincingly show that communication is the main bottleneck in the evaluated setup. For a communication-efficient method, I would expect wall-clock, throughput, or time-to-target results. Without this, it is difficult to assess the practical value of the reported communication savings.

I also think the evaluation is missing a baseline that uses a matched communication budget, which, instead of lowering payload via low-rank communication, simply synchronizes less frequently. Figure 5 studies synchronization frequency for a few ranks, but a relevant question is when low-rank communication is actually preferable to proportionally reducing the synchronization frequency.

A further concern is that some of the benefit from the full-rank QHM update may come from the more general effect of combining momentum with a fresh gradient term. Similar gradient-injection ideas (like accelerated SGD) could be tested for both low-rank and full-rank baselines. This connection is especially relevant given recent work arguing that AdEMAMix can be viewed as combining momentum with the current gradient in a similar spirit to accelerated SGD and [1].

[1]  Morwani et al. Connections between Schedule-Free Optimizers, AdEMAMix, and Accelerated SGD Variants. https://arxiv.org/abs/2502.02431

---

> ### Author Rebuttal · Authors · 2026-03-31
>
> Dear Reviewer UTAu,
>
> Thank you for your time in evaluating our work and for recognizing our technical and meaningful contributions. Please find our responses to your questions and suggestions below; please find additional figures here: https://tinyurl.com/rtucpfz3.
>
> ### **Q1: Wall-clock Comparisons**
>
> We have added an analysis of the wall-clock time implications of LoRDO versus baselines in *wallclock.pdf* for our 720M results.
>
> LoRDO achieves a significant speedup relative to DDP with low-rank optimizers due to its reduced synchronization frequency (which is set to 32 across all methods in this experiment as per our paper). We also observe that using low-rank optimizers, for the same DDP or communication-efficient regimes, incurs additional wall-clock time, as predicted by prior work [LDAdam]. We emphasize that the relatively high bandwidth (100Gpbs) setting in which we conducted our experiment obscures the  differences between DDP, MT-DAO, and LoRDO. To show the effect that bandwidth has, we provide our perplexity results for a modelled wall-clock time to show how a lower-bandwidth (10Gbps) improves our result relative to DDP (16.3% at 100 Gbps to 59.5% at 10 Gbps).
>
> Finally, we wish to emphasize that the local memory reduction of our method is equally important to the reduced communication, relative to DDP, for our work.
>
> ### **Q2: Matched Communication-Budget**
>
> Thank you for this suggestion. To match the communication budget of LoRDO (at a given rank $r$), we must reduce $K_x, K_u, K_v$ of the full-rank counterpart to match both the communication overhead of LoRDO Global and Local. We show cases where we fix $K_x = K_u =  K_v$ and where we set $K_x = 32, K_u=K_v$ to account for the reduction for both LoRDO Global and Local (see *comms_matched_experiment.pdf*)
>
> As shown in the figure, the fixed $K$ case increases the perplexity of the full-rank method relative to its baseline. Furthermore, LoRDO is superior for all ranks except $r=8$. This degradation in performance of the communication-matched baselines is due to the model parameters and momenta terms being communicated too infrequently. The increase in perplexity is lower at larger values of $r$ since the full-rank method can afford more synchronization due to the gap in the payload being smaller relative to LoRDO. As shown by prior work [DES-LOC], skewing the communication budget towards the parameters ($K_x = 32$) makes the performance hit less pronounced.
>
> Furthermore, even if the communication budgets are perfectly matched, LoRDO, under the relevant setting of $r$, still provides an up-to $8\times$ reduction for the optimizer state memory overhead.
>
> ### **Q3: Projection vs. Momentum Mixing**
>
> Thank you for this question. We have consolidated these findings (which were previously spread across the paper) into a unified experiment which we present below. Our ablations isolate the gains into two key observations:
>
> 1. **Gradient Injection To Improve Performance:** Adding the fresh gradient signal (QHM) universally improves both LoRDO and full-rank baselines. We find this to be the case throughout our work, corroborating the result of MT-DAO which shows that such an injection is crucial because it allows the momenta half-lives to safely match their synchronization intervals. Our novelty is that we reformulate the QHM update into a full- and low-rank form. This allows the QHM to have a dual use in the case of LoRDO Global, alleviating the aggregation pathologies in Section 3 in addition to improving performance.
> 2. **Global Projection Superiority:** Observing our results in Figures 2.a and b where both LoRDO-Global and Local use QHM, LoRDO-Global provides superior results, especially at lower ranks. This shows, when isolating the QHM contribution, that the global projection provides an additional improvement over the quasi-hyperbolic term.
>
> ### **Limitations**
>
> We will provide a shorted summary of our limitations in the main text of the camera-ready version.
>
> ### **Closing**
>
> We believe that these new consolidated experiments and clarifications directly address your concerns. If you are satisfied with our responses, we kindly invite you to consider raising your score.

---

> > ### Author Rebuttal · Reviewer_UTAu · 2026-04-01
> >
> > The rebuttal is helpful and clarifies several of my questions. The problem of reducing communication and memory in distributed training is clearly important. At the same time, it is still somewhat unclear how often the proposed approach would be preferred in practice over simpler alternatives, especially given the additional method complexity and tuning considerations. I will take the rebuttal into account when revisiting my score.

---

> > > ### Author Response · Authors · 2026-04-01
> > >
> > > Dear Reviewer UTAu,
> > >
> > > Thank you for your prompt response and for acknowledging that our rebuttal was **helpful** and that we have **fully resolved** the questions raised in your initial review. In light of your final comment, we would like to emphasize the following two points:
> > >
> > > - **Preference over alternatives:** We believe the results presented in our main paper, and the additional experiments presented in this rebuttal, prove that LoRDO is a well-justified and performant alternative to baselines. Specifically, LoRDO achieves **near-parity with DDP low-rank** while **requiring orders of magnitude less wall-clock time.** Additionally, it also significantly **reduces optimizer memory and communication overhead** compared to full-rank optimizers used in a communication-efficient context. These benefits make LoRDO well-suited for the memory- and bandwidth-constrained environments for which it was designed.
> > > - **On complexity and tuning:** We wish to highlight that LoRDO **introduces fewer tunable parameters** than baselines like DiLoCo, and relies on the exact same number of hyperparameters (one) as previous methods (MT-DAO). To simplify adoption, we provide **a clear methodology** for jointly tuning the learning rate and $\omega$ in Appendix G. Furthermore, we leverage CompleteP scaling guarantees to ensure that hyperparameter choices transfer robustly across model scales (Section 4.1 and App. G). While LoRDO-Global does add the complexity of communicating a global projection matrix compared to LoRDO-Local, we shown that this is crucial for the improved performance of low-rank optimizers in communication-efficient regimes. Ultimately, we offer principled and scalable guidance for deriving performant settings in LoRDO.
> > >
> > > We hope you will consider these two points alongside our full rebuttal when adjusting your score.

---

### Official Review · Reviewer_4U6k · 2026-03-09

**Soundness:** 2
**Presentation:** 3
**Significance:** 2
**Originality:** 3
**Overall Recommendation:** 3
**Confidence:** 4

**Summary:**

This paper studies how to combine low-rank adaptive optimizers (GaLore, LDAdam) with infrequent-communication distributed training (DiLoCo, DES-LOC). The authors identify two failure modes of naively applying these techniques to the communication-efficient setting: noisy local projections and stagnant global projections. As a solution, they propose LoRDO, which uses a global projection to maintain stability while also leveraging full-rank quasi-hyperbolic updates to avoid only optimizing within a low-rank subspace. Experiments on decoder-only transformers up to 720M with 4 workers show near-parity with synchronous low-rank DDP at roughly 10x less communication.

**Compliance With Llm Reviewing Policy:**

Affirmed.

**Final Justification:**

See reviewer-reviewer discussion.

**Key Questions For Authors:**

- Does the algorithm converge?
- Is a Muon inner optimizer compatible with LoRDO?
- Does CompleteP trivially apply to infrequent-communication distributed training (e.g., DiLoCo, DES-LOC)?

**Limitations:**

Yes

**Strengths And Weaknesses:**

**Strengths**
- The core strength of this paper is the observation that a shared global projection permanently traps the aggregated pseudo-gradient in a fixed rank-r subspace, and the proposed QHM, which is shown to recover the resulting performance degradation relative to low-rank ddp without increasing memory overhead.
- I find the writing to be clear and to the point.
- I like the presentation of the results section in terms of research questions, making it easy to map findings to sections.
- The method matches the performance of the DDP baselines while reducing communication.
- I like the author’s idea to leverage completeP in order to reduce the cost of their hyperparameter search at larger scales.

**Weaknesses**

- **Memory savings overclaimed** On line 213, column 2, the authors provide the memory savings of their method, <3X. However, on line 430 of the appendix, the authors state “Furthermore, it reduces optimizer memory and communication costs by up to 12×”. Moreover,  Communication-efficient optimizers perform best when using an outer momentum [1,2]. This requires two additional accumulators of size O(pq): (1) the outer momentum, (2) the previous parameters $\bar{x}^m_{t-K_x}$. Even when using averaging as the outer optimizer, LoRDO must store an additional state ($\bar{x}^m_{t-K_x}$) which is not reported on line 213.



- **Practical Memory and communication savings and Comparing to Prior work** Since LoRDO synchronizes optimizer states as in DES-LOC [1] and synchronizes the full pseudogradient (line 20 of Algorithm 1). It is trivially less communication-efficient than DiLoCo [2] or MuLoCo [2]. Moreover, LoRDO’s use of a full-rank error feedback accumulator makes it less memory efficient than MuLoCo [2] (published previous work). We assume that the authors made this choice because DiLoCo and MuLoCo have not theoretically been shown to converge. However, given the practical appeal of these algorithms and the demonstrated superior performance of MuLoCo to baselines that match LoRDO, a discussion is warranted.

- **Proof of convergence** I miss a proof of convergence or an explanation of why this is trivial from [1].

- **Hyperparameters** CompleteP was designed for data-parallel training, and it has not been shown to work for infrequent-communication distributed training (DiLoCo, DES-LOC). Could the authors explain what steps they took to ensure the parameterization effectively transfers optimal hyperparameters in this case?



**Typos & nit picks**
- Figure 1 (a) and (b) have the same subcaption without any indication of the difference between both figures.



**References**
---
[1]DES-LOC: Desynced Low Communication Adaptive Optimizers for Training Foundation Models; https://openreview.net/forum?id=6N2qFixxYZ]

[2][DiLoCo: Distributed Low-Communication Training of Language Models; https://arxiv.org/abs/2311.08105]

[3][MuLoCo: Muon is a practical inner optimizer for DiLoCo; https://icml.cc/virtual/2025/51842]

---

> ### Author Rebuttal · Authors · 2026-03-31
>
> Dear Reviewer 4U6k,
>
> Thank you for your time and effort in reviewing our paper, and for identifying that our work is both original and clear in its presentation. Below, we address your specific concerns.
>
> ### **Q1: Memory Overhead Claims**
>
> We respectfully disagree that our memory savings are overclaimed. The line on 213 refers to the total combination (model + projection matrix +  optimizer states for memory, and pseudogradient + projection matrix + optimizer states for communication). In the text, however, we focus on the optimizer states, following prior works like GaLore and LDAdam. In this case, the savings depend entirely on the choice of rank $r$. All of our references to $8\times$ or $12\times$ refer to the optimizer memory and communication cost, as explicitly stated in the text (”reduces **optimizer** memory and communication costs by up to 12×”). As per our response to Reviewer 4E3W, choosing $p=q=768$ and $r=64$ yields a 12x reduction.
>
> Additionally, in single program multiple data setups (SPMD), **local workers do not need to store the pseudogradient**. The workers only need to communicate their current model copy at the synchronization interval, where the pseudogradient is computed by the global orchestrator. The average model can then be deleted. Finally, as presented to Reviewer 4E3W, the outer optimizer complements the local optimization procedure; LoRDO provides improved performance over DiLoCo without the need for an additional server momentum.
>
> ### **Q2: Comparison to DiLoCo**
>
> You correctly identify that we communicate optimizer states based on prior work showing this is necessary for stability, performance, and convergence guarantees. However, LoRDO is designed flexibly to accommodate two critical factors:
>
> 1. **Optimizer State Sync:** Practitioners can tune $K_x, K_u,$ and $K_v$ based on their specific communication constraints.
> 2. **Outer Optimizer:** The choice of outer optimizer can be flexibly selected, as it is complementary to the local optimization design (our primary focus).
>
> Please see our discussion with Reviewer 4E3W on comparisons with DiLoCo. Regarding MuLoCo, we did not compare to it as it was presented at a **non-archival workshop,** to the best of our knowledge.
>
> ### **Q3: CompleteP Transferability**
>
> CompleteP ensures hyperparameters transfer across model sizes by appropriately scaling the initial weights and their corresponding updates to maintain consistent feature learning. Since the magnitude of these weight updates is strictly controlled, this scales predictably regardless of synchronization frequency, ensuring these transfer conditions hold even in infrequent communication regimes. This aligns with prior work that also utilizes the CompleteP framework [MT-DAO]. We empirically validated this: the optimal learning rate found via sweeping at 16M (App G) remained stable when checked in the immediate neighborhood at 125M and 720M. We omitted further sweeps due to the aforementioned theoretical grounding.
>
> ### **Q4: Convergence**
>
> We have derived a preliminary theoretical convergence sketch for LoRDO (for the smooth non-convex setting with first order momentum), and we plan to include it in the final version once verified. The analysis combines steps from LDAdam (for handling low-dimensional subspace transitions and error feedback) and from MT-DAO (to address multiple local steps and the QHM structure of the main update). The first step is to define virtual iterates $z_t$ that follow (preconditioned) SGD.
>
> $z_{t+1} = z_t - \eta G_t.$
>
> For the full-rank QHM case specifically, we can achieve this by setting
>
> $z_t = \omega ( \frac{1}{1-\beta} x_t^u - \frac{\beta}{1-\beta} x_{t-1}^u ) + (1-\omega) x_t^g - \eta \omega E_{t-1},$
>
> where $x_{t+1}^g = x_t^g - \eta G_t$ is updated using only $G_t$, and $x_{t+1}^u = x_t^u - \eta Q_t \bar{u}_t$ is updated using $Q_t \bar{u}_t$. Once such a virtual sequence is established, Step 2 (smoothness over virtual iterates) and Step 3 (one-step progress) of the MT-DAO analysis follow directly. The remaining task is to bound the client drift term $\||x_t - x_t^m\||^2$ and the virtual drift term $\||z_t - x_t\||^2$, in order to derive guarantees with respect to the averaged model $x_t$ rather than the virtual iterate $z_t$. Bounding these terms should be feasible by leveraging the error feedback analysis developed for LDAdam.
>
> ### **Q5: Compatibility with Muon**
>
> Muon is compatible with LoRDO by either: i) additionally down-projecting the gradient prior to momentum calculation, or ii) using a low-rank calculation similar to Dion where LoRDO would serve as the communication-efficient training generalization. We leave full empirical investigation of this for future work.
>
> ### **Presentation**
>
> The top subfigure of Fig. 2 will be labeled 16M in the camera-ready version.
>
> ### **Closing**
>
> We believe these clarifications directly address your concerns. If you are satisfied with our responses, we kindly invite you to consider raising your score.

---

> > ### Author Rebuttal · Reviewer_4U6k · 2026-04-03
> >
> > **Regarding memory overhead** Given the additional experiments provided in the rebuttal, finding that LoRDO using averaging as the outer optimizer outperforms Nesterov momentum, my concern regarding the reporting of memory overhead is resolved.
> >
> > **On outer Nesterov hurting performance for LoRDO** This is an interesting finding given the strong performance of the Nesterov outer optimizer reported in [1]. Did the authors additionally tune the server learning rate beyond 0.2,0.4 in the “lordo_outer_opt.pdf” experiment? Moreover, was the inner learning rate also tuned alongside these outer hyperparameters?
> >
> > **LoRDO outperforms full rank DiLoCo in 125m_diloco_comparison.pdf** This is quite an interesting result and should be highlighted in the main manuscript!
> >
> > **Regarding the use of CompleteP** I looked through App G, but found three figures reporting the hyperparameter sweep for 16M. I imagine that some of these figures refer to different model sizes?  I also looked through the provided reference [MT-DAO], but did not find a justification for why CompleteP remains valid under different choices of outer optimizer (e.g., Nesterov and averaging) and when using low-rank inner optimizers.
> >
> > **Compatibility with Muon/Comparison to MuLoCo** Even if no explicit experimental comparison is made, it would be good to comment on the compatibility of LoRDO with these methods in a future work section.
> >
> > The authors have resolved my main concerns regarding convergence and memory overhead. That being said, I will wait to hear back from the authors regarding my additional questions before updating my score.

---

> > > ### Author Response · Authors · 2026-04-06
> > >
> > > Dear Reviewer 4U6k,
> > >
> > > Thank you for finding our rebuttal satisfactory in answering some of your concerns. We respond to your additional questions below:
> > >
> > > ### Q1: Nesterov Outer Optimizer
> > >
> > > We did tune the server learning rate over the full grid $\{0.2, 0.4, 0.6, 0.8, 1.0\}$ as per "Scaling Laws for DiLoCo". We also considered different learning rates, beta values for the local optimizer (we noticed an interaction with the server momentum) and found no values that successfully transferred from the originally recommended settings in [1]. In total, we have run $\approx 430$ tuning experiments, where we initially omitted these results for readability. Please see the updated `lordo_outer_opt.pdf` and the new figure `beta_outer_opt.pdf`. While exploring all possible interactions between the inner and outer optimizers warrants further investigation, an exhaustive ablation is outside the core scope of this work (as highlighted by Reviewer 4E3W).
> > >
> > > ### Q2: CompleteP Justification
> > >
> > > We note that, as per Tensor Programs V (Sec. 3-5), we must ensure that hidden pre-activations are model-scale invariant. Specifically, if $h=Wx$ (where $W \in \mathbb{R}^{d \times d}$ and $x \in \mathbb{R}^d$), the change in pre-activations after an update step $\Delta W$ is:
> > >
> > > $$\Delta h=(\Delta W)x$$
> > >
> > > CompleteP-like frameworks ensure that as $d \to \infty$, the magnitude of this update remains bounded with increasing model size:
> > >
> > > $$\Delta h=\Theta(1)$$
> > >
> > > **Using low-rank optimizers:** Low-rank optimizers project updates onto a lower-dimensional subspace of rank $r$. In this regime, the magnitude of the low-rank update, relative to a full-rank update, shrinks by an $r$-dependent factor:
> > >
> > > $$\Delta W_{LR}=\sqrt{\frac{r}{d}} \Delta W_{full}$$
> > >
> > > As such, the change in pre-activations under the low-rank optimizer becomes:
> > >
> > > $$\Delta h_{LR}=\left( \sqrt{\frac{r}{d}} \Delta W_{full} \right) x = \sqrt{\frac{r}{d}} (\Delta h_{full})$$
> > >
> > > This could potentially violate the CompleteP requirement of $\Delta h=\Theta(1)$ if $r$ were frozen as $d$ increased with model scale. However, we tuned the learning rates using a fixed width-to-rank ratio for the low-rank optimizer versions across all model scales. We define this as a constant $\rho$:
> > >
> > > $$\frac{d}{r}=\rho \implies \sqrt{\frac{r}{d}} = \frac{1}{\sqrt{\rho}}$$
> > >
> > > Applying this constant into the pre-activation update equation yields:
> > >
> > > $$\Delta h_{LR}=\frac{1}{\sqrt{\rho}} (\Delta h_{full})$$
> > >
> > > Since $1/\sqrt{\rho}$ is a positive, width-independent constant, it follows that:
> > >
> > > $$\Delta h_{LR}=\frac{1}{\sqrt{\rho}} \Theta(1) = \Theta(1)$$
> > >
> > > This proves that low-rank optimizers preserve the scaling for CompleteP. The scalar $1/\sqrt{\rho}$ acts as a global multiplier that is absorbed into the base learning rate.
> > >
> > > **In communication-efficient settings:** After $K$ steps, the local worker model $m$ can be expressed as the original weights plus the accumulated local updates:
> > >
> > > $$W_m=W_0+\Delta W_m$$
> > >
> > > When the $M$ local models are aggregated at the communication step, the new global weights are obtained via averaging:
> > >
> > > $$W_{new}=\frac{1}{M} \sum_{m=1}^{M} W_m$$
> > >
> > > $$W_{new}=\frac{1}{M} \sum_{m=1}^{M} (W_0+\Delta W_m) = W_0 + \frac{1}{M} \sum_{m=1}^{M} \Delta W_m$$
> > >
> > > From this, the effective global update is $\Delta W_{global}=\frac{1}{M} \sum_{m=1}^{M} \Delta W_m$. To prove that hyperparameter transfer holds, we must show that the global change in pre-activations ($\Delta h_{global}$) remains model-scale invariant. We can express this global feature update as:
> > >
> > > $$\Delta h_{global}=(\Delta W_{global})x = \left( \frac{1}{M} \sum_{m=1}^{M} \Delta W_m \right) x = \frac{1}{M} \sum_{m=1}^{M} (\Delta W_m x) = \frac{1}{M} \sum_{m=1}^{M} \Delta h_m$$
> > >
> > > Because the number of workers $M$ is a finite constant independent of the model width $d$, and because CompleteP guarantees that every local update satisfies $\Delta h_m=\Theta(1)$, it mathematically follows that:
> > >
> > > $$\Delta h_{global}=\frac{1}{M} \sum_{m=1}^{M} \Theta(1) = \Theta(1)$$
> > >
> > > This reflects the linear combination primitive in Tensor Programs III. Averaging is strictly a linear operation over width-independent combinations; no width-dependent scaling factors are introduced or destroyed. If CompleteP dictates a specific $1/d$ scaling for a layer's update magnitude to achieve $\Theta(1)$ feature learning, the aggregated update strictly preserves this limit, guaranteeing zero-shot transfer without modification. The case for depth scaling is similar.
> > >
> > > Proving CompleteP for every local and outer-optimizer combination is beyond this paper's scope. Nevertheless, these mathematical arguments support our methodology, showing the theory transfers well.
> > >
> > > ### Closing
> > >
> > > We believe these clarifications directly address your concerns. As suggested, we will include the new DiLoCo comparisons, outer optimizer experiments, and discussions on compatibility with Muon in the camera-ready version. If you are satisfied with our responses, we kindly invite you to consider raising your score.

---

### Official Review · Reviewer_3rHy · 2026-03-12

**Soundness:** 2
**Presentation:** 2
**Significance:** 2
**Originality:** 3
**Overall Recommendation:** 3
**Confidence:** 4

**Summary:**

This paper presents LoRDO to improve memory and bandwidth bottlenecks in distributed training. Specifically, they propose replacing the inner optimizer of local-update methods with low-rank optimizers which has the benefit of reduced memory costs, and in case of MT-DAO/DESLOC, it improves communication costs.

**Compliance With Llm Reviewing Policy:**

Affirmed.

**Final Justification:**

After considering the rebuttal and the added experiments, I am maintaining my score at 3. The paper has clear technical merits, and the proposed method is interesting with practical value. I appreciate the clarification regarding memory overhead and the new experiments. However, my main concerns regarding the main motivation of the paper remain:

1. The naive combination introduced in L98-104 (also referred to as LoRDO-Local + no-QHM) is not properly investigated. This method does not introduce an additional HP $ \omega $, and it is very important to establish the impact of QHM for LoRDO-Local as $ \omega $ affects the learning rate (Section G). However, the new experiment only considered LoRDO-local with QHM.

2. The gap between the local variants and LoRDO-global is not properly investigated. The local versions are highly dependent on the local batch size, and the primary experiments in the paper use a single, small batch size and model size for HP-tuning. The new experiments (`2x batch.pdf` and `K_vs_BM.pdf`) are interesting and helpful; however, how the hyper-parameters were set for these experiments was not clarified, and if the same HP from lower batch sizes are used, it makes the new experiments difficult to rely on. Furthermore, the K_vs_BM experiment still uses a very small local token horizon for M>1, and the point of this experiment for me was to fix $K \times BS$ (for the same M, while increasing/decreasing K and BS), which means the same token budget per outer step, and find the critical batch size for each method before comparing local variants against LoRDO-Global. Therefore, since local variants, and especially the naive combination, heavily depending on local gradients, whether these methods are underperforming due to an unoptimized BS still remains unclear. I believe this concern requires LoRDO to investigate batch size more carefully than most baselines.

**Key Questions For Authors:**

See weaknesses.

It would be nice if authors could summarize the HP search spaces used for each method and their optimal values in a table.

**Limitations:**

yes

**Strengths And Weaknesses:**

### Strengths

The idea is novel and interesting with practical use case, and authors show strong performance compared to DDP and low-rank/local-step baselines. Also, the naive global combination failure is interesting and properly discussed.

---

### Weaknesses

1. The proposed method, especially LoRDO-Global, is not fully justified, especially as it introduces a new hyperparameter search space for $\omega$ and affects learning rate.
  - One of the main motivations for LoRDO-Global is that the local variants have a significantly lower token context for updating the projection matrix Q. This suggests that as H increases, one should expect LoRDO-Global to outperform the local variants even more than lower H. However, Figure 5 shows the opposite for both r=8 and r=128. Here, we see Local QHM significantly outperforming Global QHM as H increases, and at r=128, the naive combination outperforms both.

  - Especially, 16M model is extremely small and uses a significantly small batch size per worker $B_{local}=8$ (assuming Figure 5 is done using a 16M model). The signal would become stronger with higher local batch sizes. Authors should consider a thorough ablation by fixing $H \times B_{local}$, e.g., by reducing H and increasing B, to enhance the local signal without modifying the pseudo-gradient token size.

2. The discussion in Sec 5.5 and the information in Tables 2 and 3 are highly misleading. In Sec 5.5, the authors mention that optimizer state memory and communication overhead are reduced by 8x. While correct, this does not matter as there is a communication overhead associated with low-rank projections:

- Regarding communications, LoRDO is in fact more **inefficient** than DiLoCo/MuLoCo and much more than sota communication efficient methods such as SparseLoCo.

- Regarding memory, including the model parameters memory, LoRDO-Global is $O(2pq+4rq)$ while DiLoCo/SparseLoCo are $O(3pq)$, and MuLoCo $O(2pq)$. This is further amortized as practical large-scale training involves sharding distributed strategies.

3. In L761-764, the authors claim that aggregating M pseudo-gradients of rank r recovers a representation in the original space. This is clearly not the case (e.g., if $M \times r < d$). Could you please clarify?

4. The scale of experiments is limited. Most ablations are done on a extremely small model with a very low batch size. The largest model considered is also relatively small, especially as they do not further tune important hyperparameters and `Local non-QHM` is not included.

5. The writing overall could be further polished to improve readability. Moreover, the rank used in Figure 4 is not clarified, and all figures mention model sizes except for Fig 5. Also, in Table 3 left column, 3 rows repeat twice. The proposed QHM has two variants (low-rank, full-rank), but it's not clarified which is used in the main text (especially for lordo-local), and only in the appendix L1329-1330 is this clarified. I believe this should be in the main text or captions.

---

> ### Author Rebuttal · Authors · 2026-03-31
>
> Dear Reviewer 3rHy,
>
> Thank you for reviewing our paper and for recognizing that our work presents novel, interesting ideas with highly practical use cases. Below, we address your specific questions and feedback; please find additional figures here: https://tinyurl.com/rtucpfz3.
>
> ### **Q1: LoRDO Global**
>
> - **Hyperparameters:** We respectfully disagree that introducing a new hyperparameter makes our method ill-justified. We introduce fewer hyperparameters (one) than DiLoCo (two), and we provide a clear methodology to jointly tune the learning rate and $\omega$ in Appendix G. Furthermore, leveraging CompleteP scaling guarantees that hyperparameter choices transfer robustly across model scales (Section 4.1 and App. G).
> - **Global vs Local**: Prior work [DES-LOC] has shown that aggregating the model parameters highly infrequently is detrimental for stability and machine learning performance (where H>128 in Figure 5) as the model replicas diverge too far, introducing noise and degrading the quality of the pseudogradient. As such, this explains why the local method, which depends only on the instantaneous gradient, is less affected in these regimes than the global counterpart that uses the now low-quality pseudogradient. However, in regimes where $H$ is well selected ($H \leq 128$), this does not occur. Furthermore, as stated in our paper, LoRDO-Global is best at lower ranks; local outperforms at $r=128$ as predicted by our derivations; see Section 5.2.
> - **Projection Dependence on H:** For the 16M model, our local batch is 16 (equal to to our DDP per-worker batch size) for a global batch size of 64 which is in-line with the critical batch size expected for models of this scale [An Empirical Model of Large-Batch Training, How Does Critical Batch Size Scale in Pre-training?]. Finally, Figure 6 already provides an investigation into this point as $H$ was frozen throughout.
>
> ### **Q2: Communication and Memory Overhead**
>
> - **Memory:** LoRDO Global operates at $O(pq + pr + 2rq)$ as we sum error feedback with the gradient (as per LDAdam) and discard the low-rank gradient after updating momenta. Our overhead is strictly less than DiLoCo ($O(3pq)$) and does not require storing server momentum of size $pq$. It is also less than MuLoCo ($O(2pq)$) whenever $r<\frac{p}{3}$, which holds for most of our experiments.
> - **Momenta Aggregation/Outer Optimizers:** Communicating optimizer states offers a trade-off: higher overhead for guaranteed convergence, stability and improved performance ({*16m/125m}_diloco_comparison.pdf and opt_state_sync.pdf*). LoRDO allows for the practitioners to toggle optimizer state aggregation based on constraints. We used averaging as the most principled base case, however, we view the outer optimizer to be complementary to the local optimizer design. Please see further discussion with Reviewer 4E3W.
> - **Further Compression to Pseudogradient:** We note in the related work that SparseLoCo’s modifications to pseudogradient communication are strictly orthogonal to our contributions, and would only further improve our communication overhead.
> - **Prior Work:** Finally, we did not explicitly compare to MuLoCo nor SparseLoCo since they have only been presented at **non-archival** workshops, to the best of our knowledge.
>
> ### **Q3: Full Space Exploration**
>
> We will note in the Appendix that aggregated rank capacity can exceed the bottleneck ($M \times r \ge \min(p, q)$) when sufficiently many workers are used. However, the core mechanism holds: full-rank QHM prevents LoRDO-Global from being trapped in a rank-$r$ subspace. By temporally accumulating $M \times r$ solutions and transitioning across subspaces, it spans the full parameter space—the exact temporal rank recovery mechanism used by GaLore and LDAdam.
>
> ### **Q4: Experimental Validity & Scale**
>
> While we would love to extend our experiments to larger scales, billion-parameter runs are beyond our compute and time constraints for this short rebuttal window. We respectfully disagree that our evaluation scale is limited. We scaled batch sizes appropriately for each model size, strictly following "Benchmarking Optimizers for Large Language Model Pretraining”. We maintain that our comprehensive 720M parameter experiments provide robust, sufficient evidence that our method scales effectively, and we expect this to hold at larger sizes.
>
> ### **Presentation Updates**
>
> We will correct the following in the camera-ready.
>
> - Fig 4: Note rank $r=8$; Fig 5a: Note 16M model size.
> - Table 3: Clarify duplicate rows show decoupling of pseudogradient into projection and accumulated over time.
> - Main text: Explicitly state QHM variants for Local/Global.
> - Table 4: Integrate learning rate and $\omega$ values from App G.
>
> ### **Closing**
>
> We believe these clarifications directly address your concerns regarding our hyperparameters, communication overhead, and experimental scale. If you are satisfied with our responses, we kindly invite you to consider raising your score.

---

> > ### Author Rebuttal · Reviewer_3rHy · 2026-04-04
> >
> > Thanks to the authors for the detailed rebuttal.
> > 1. First, the main ablation study regarding the primary motivation for the paper is still missing. My concern was not whether a global batch size of 64 is reasonable for the 16M model size. My main concern is as the local variants, especially the naive combination, depend heavily on the local gradients (L160-163), the small batch size of 32.8k tokens per worker may be insufficient. Therefore, an experiment with tuned local variants at a higher per-worker token horizon (eg, at 125M scale with slightly higher than critical batch size) is essential to motivate the proposed method, especially as LoRDO is primarily proposed for large-scale training. Furthermore, an experiment with fixed $K \times BS$ with different combinations of K and BS for local variants would be very helpful to further support the necessity for the global version. In other words, while using critical batch size is an acceptable standard in many studies, the dependency of local variants on local batch size requires further exploration.
> >   - Regarding the rebuttal's comments on this point, my mention of the new hyper parameter was in this context, as the naive combination avoids $ \omega $ (and not as compared to DiLoCo). Also, the paper uses CompleteP to transfer the learning rate, but not $ \omega $, so the rebuttal's statement is stronger than Appendix G. I also believe authors meant to refer to Figure 4 for the batch-size discussion. I would like to point out that this figure still uses the small 16M model size and largest local batch size has only 1 worker, and the figure is missing naive combination, so it is still difficult to draw any conclusions using this Figure. Finally, I appreciate the authors noticing I mistakenly used H instead of K in my review.
> >
> > 2. Regarding Figure 5, I appreciate authors clarification that at extremely high K>128, the degradation of LoRDO-global is due to quality degradation of pseudo-gradients. I believe a brief explanation of this in the caption or in Section 5.4 would greatly clarify this experiment as it is unclear why K>128 was even considered.
> >
> > 3. I appreciate the clarification on the memory. I agree that incorporating the gradients, LoRDO has less memory overhead than baselines and this part of my concern was properly addressed. However, the rebuttal mentions $O(pq+pr+2rq)$ whereas table 3 shows $O(pq+pr+3rq)$ as the overhead of LoRDO-Global/Local + Full-Rank QHM. Could the authors clarify which is correct?
> >
> > (minor) 4. On full-space exploration, even with high number of workers, true full-rank recovery is not guaranteed without sufficient diversity across worker subspaces, so while the rebuttal's phrasing is still too strong.
> >
> > While some of my concerns have been properly addressed in the rebuttal, the main concern regarding the main motivation of the paper still remains, and I look forward to authors response before updating my score.

---

> > > ### Author Response · Authors · 2026-04-06
> > >
> > > Dear Reviewer 3rHy,
> > >
> > > Thank you for engaging with our rebuttal and providing further constructive feedback. Please find our responses to your remaining points below:
> > >
> > > **Q1: LoRDO-Global vs. Local & Batch Size Ablations**
> > >
> > > We agree that the local variant's reliance on instantaneous gradients is key, which is precisely why we hypothesized it would be a deficient choice in the communication-efficient regimes we study. While using the critical batch size as the global batch size is standard practice in LLM training, scaling the number of workers in these regimes, by construction, naturally divides this global batch into smaller local batch sizes.
> > >
> > > To directly address your concern, we conducted a new experiment at the 125M scale across all ranks we initially considered using a 2x global batch size (resulting in a 2x local batch size per worker). We halved the training steps to ensure an iso-token comparison (please see the attached `2x_batch.pdf`). As you intuitively predicted, increasing the local batch size does improve the performance of the local variant (LoRDO-Local). **However**, **LoRDO-Global still maintains its superiority** (particularly at small ranks as before), and it improves on perfromance relative to the local method even when its global batch size is kept constant. This supports our central claim: in communication-efficient training, using a global projection matrix derived from aggregated pseudo-gradients is generally superior to relying on local projection matrices, even when pushing the global batch size beyond the critical threshold.
> > >
> > > Furthermore, we have provided an additional ablation mapping the interaction between $K$ and $B_{\text{Local}} \times M$ for 16M models across a single rank ($r=8$), expanding our initial Figure 8 experiment (please see `K_vs_BM.pdf`). This explores the local batch size interacting with the number of steps. The results show that in all cases where $K > 1$ (the regimes that are communication-efficient relative to DDP), LoRDO-Global consistently outperforms the local method. While fully mapping the continuum between $K$ and local batch size is an open problem for future work, these new experiments confirm that within the regimes we have considered, the global projection remains necessary to provide superior performance.
> > >
> > > **Regarding $\omega$ and CompleteP***:* We wish to clarify that the "naive" method (LoRDO-Local) does utilize the additional parameter $\omega$. As discussed with Reviewer UTAu, $\omega$ is necessary to allow the momenta half-lives to safely match their synchronization intervals. Therefore, the only difference between LoRDO-Global and LoRDO-Local is how the projection matrix is derived. Additionally, we do use CompleteP to transfer both the learning rate and $\omega$, as optimizer-related hyperparameters are transferable per Tensor Programs V. We apologize if Appendix G was ambiguous on this front; we will make this explicit in the final text.
> > >
> > > **Q2: On the Choice of $K > 128$**
> > >
> > > We appreciate the suggestion to clarify this in the text. In the camera-ready version, we will add the following explanation to supplement our discussion. While $K > 128$ is suboptimal from a pure optimization perspective, such extreme values are sometimes explored in related literature (e.g., original DiLoCo and DES-LOC) to push the limits of communication savings and reduce wall-clock time relative to DDP. We included these data points to provide a complete picture of performance degradation at these limits.
> > >
> > > **Q3: Clarification on Memory Overhead**
> > >
> > > The $\mathcal{O}(pq + pr + 2rq)$ cost stated **in the text and our rebuttal** is correct for steady-state memory. The additional $rq$ component shown in Table 3 accounts for the low-rank gradient; however, this is a tensor that can be immediately freed from memory once applied to the momenta. Table 3 was initially constructed to show all interacting components comprehensively, without accounting for such optimizations. In the camera-ready version, we will update Table 3 to reflect the optimized $\mathcal{O}(pq + pr + 2rq)$ overhead to ensure strict consistency with the main text.
> > >
> > > **Q4: Full-Space Exploration**
> > >
> > > We appreciate this nuanced point. Our claim regarding full subspace exploration relies not only on the number of workers but also on the temporal component—reflecting the mechanisms in LDAdam and GaLore. This temporal component is what allows the low-rank updates to effectively match the performance of their full-rank counterparts in certain cases. However, we will expand upon this explanation in the camera-ready version to accurately reflect this distinction and avoid any further ambiguity.
> > >
> > > **Closing**
> > >
> > > We believe these new experiments and clarifications directly address your remaining concerns regarding the global vs. local motivation, memory overhead, and hyperparameter usage. If you feel these updates strengthen the paper and resolve your concerns, we kindly ask you to consider raising your score.

---

### Official Review · Reviewer_4E3W · 2026-03-15

**Soundness:** 4
**Presentation:** 4
**Significance:** 3
**Originality:** 3
**Overall Recommendation:** 5
**Confidence:** 4

**Summary:**

The paper explores distributed optimization methods that use local updates with periodic synchronization in order to reduce communication, specific building upon DES-LOC which averages pseudo-gradients and local optimizer states. Upon this they consider having the local optimizer be low rank, in LDAdam (which  builds upon Galore) style. However they note that naïve integrate would have the projection matrix be calculated from a local mini-batch and not the entire global batch, which causes problems (they give theoretical reasons for this and verify it experimentally). Thus they propose to compute global projection matrices using the outer pseudo-gradient (a nice idea that requires no extra communication), but then note that this theoretically would restrict updates to a fixed subspace. Thus they propose to use Quasi-Hyperbolic Momenta in their updates, which has both a momentum term that is low rank and a gradient term that is high rank, allowing full rank exploration.

**Compliance With Llm Reviewing Policy:**

Affirmed.

**Final Justification:**

The authors have properly addressed all my concerns, most importantly that they properly compared to DiLoCo and even found that Nesterov momentum does not help in their situation (which is very interesting/surprising). I, like many of the reviewers, found their explanation of the communication savings in the submission to be quite confusing, but they have now properly addressed it (and I hope they properly integrate this feedback in their revised version).

I overall found this work to be really solid: well written and planned out with properly investigations about their hypotheses that even disentangle them properly (not just "the method worked"). Thus I am maintain my positive score of 5.

**Key Questions For Authors:**

Given in strengths and weaknesses, please especially address the questions about communication savings and about comparison to DiLoCo.

**Limitations:**

Discussed in the appendix.

**Strengths And Weaknesses:**

The main idea of this work is to build upon DES-LOC which also synchronizes optimizer states, and makes optimizer states less memory heavy in a way that doesn't suffer from low local batch size or restricted sub-spaces. This reduces the communication cost of the optimizer states, but doesn't reduce the communication cost of the pseduo-gradient. It also reduces the memory size needed on each worker since the optimizer states are smaller.

The paper is very well written and planned out. They propose to reduce the optimizer states using a standard approach (LDAdam), note this causes an issue (unstable projection from low batch info) and propose a fix (globally derived projection), and then note this causes an issue (fixed subspace) and fix it (QHM). The issues and fixes are both properly explained theoretically. What's also really, really nice is that they properly test these issues and fixes. It's pretty hard to properly disentangle the benefits of the two fixes, but they do this well, they compare local vs global to show the second issue, and then change local batch sizes to show the first. They also investigate quantities that are related to their hypothesis, such as basis change and stable rank+spectral gap.

In section 3.2, multiple times it says that the full-rank QHM injects a full-rank signal into the pseudo-gradient, but its actually doing more than that right? It's injecting it into the local update (which of course then affects the pseudo-gradient) so the local workers are taking a combination of a low-rank and a full-rank step. I feel like the wording in that section makes it sound like full-rank exploration is only happening in the outer optimization step.

Figure 2 is not properly explained, it gives two plots on top of each other with the same sub-caption and no explanation anywhere else and what the difference between them is.

One thing thats really interesting is that the trick of using the global pseudo-gradient, used to get global information for computing a global projection matrix, works well, and as discussed in Sec E.3, seems to be a better choice than the global (1-step) gradient. Some intuition is given for this (better curvature information), but that suggests that the standard low-rank projection framework can be improved with second-order information: I'm curious if this is the case (is there's any information or proper arguments for this?).

I'm confused about the claims about how much the communication is reduced, it says 8x, 10x, and 12x in different places. I would have thought it would be <3x, because you go from pseduo-gradient + 1st moment + 2nd moment to pseduo-gradient + low rank 1st moment + low rank 2nd moment.

I think the paper really should compare to DiLoCo because it has even lower communication volume: pseudo-gradients only, not inner optimizer states. DiLoCo supposedly stabilizes optimization by having Nesterov momentum as the outer optimizer, which newer work seems to indicate can properly handle the issues of pseudo-gradients ([1]). Since the propsoed work has the additional communication of (heavily reduced) optimizer states, it would be good to see if the additional optimizer state communication improves the final performance of optimization, or perhaps makes optimization more stable and allows more steps between synchronization.

[1] Kallusky, Dominik & Rao, Vinay & Nandavanam, Vishal & Shi, Hao-Jun. (2025). SNOO: Step-K Nesterov Outer Optimizer - The Surprising Effectiveness of Nesterov Momentum Applied to Pseudo-Gradients. 10.48550/arXiv.2510.15830.

Furthermore, in discussing Nesterov momentum, the QHM paper shows that Nesterov momentum is a special case of this, I wonder if there's any useful connection that can be used in this work.

---

> ### Author Rebuttal · Authors · 2026-03-31
>
> Dear Reviewer 4E3W,
>
> We thank you for your time in evaluating our work and for recognizing its experimental and theoretical quality. Below, we address your specific questions and comments; please find additional figures here: https://tinyurl.com/rtucpfz3.
>
> ### **Q1: Clarification on Communication Costs**
>
> The $3\times$ term refers to the total payload (model + projection matrix +  optimizer states for memory, and model parameters/pseudogradient + projection matrix + optimizer states for communication). However, in the text, we focus on the *optimizer states*, following prior works like GaLore and LDAdam. In this case, the savings depend entirely on the choice of rank $r$. All of our references to $8\times$ or $12\times$ refer to fact that LoRDO reduces the optimizer memory and communication cost, as explicitly stated in the text (”reduces **optimizer** memory and communication costs by up to 12×”).
>
> - **Optimizer State Communication:** When comparing Adam to low-rank optimizers (LoRDO-Local/Global) in communication-efficient training, the primary reduction comes from the momenta states. Assuming $K_u = K_v$, using Adam communicates a payload of $2pq$, whereas low-rank optimizers communicate $2rq$. Using $p=q=2048$ and $r=256$ (from our 720M experiment), the *memory and communication improvement for optimizer states* is $\frac{2pq}{2rq} = 8\times$, explaining our stated improvement. In the case of the 125M model experiments $p=q=768, r=64 = 12\times$.
> - **Overall Communication vs. DDP:** Relative to DDP using a low-rank optimizer, the overall communication benefit for LoRDO Global is $(\frac{1 + \frac{r}{q}}{K_x} + \frac{1}{K_u} + \frac{1}{K_v})^{-1}$. This accounts for the reduced communication relative to DDP, the r**educed size of the momenta**, and the **added cost of the global projection matrix** in LoRDO-Global. Setting $K_x=K_u=K_v=32$, $r=256$, and $q=2048$ yields a $\approx$ 10x total communication improvement.
>
> We will update the main text to make these elements clearer.
>
> ### **Q2: Comparison to DiLoCo**
>
> We view modifications to the outer optimizer (like DiLoCo’s use of Nesterov momentum) as complementary to the design of the local optimization procedure. We chose averaging as our baseline because it is the most general case of LoRDO, where we can accomodate any choice of outer optimizer. To directly compare to DiLoCo in {*16m/125m}_diloco_comparison*.pdf, we made two adjustments: disabled momenta synchronization ($K_u = K_v \to \infty$) and set the outer optimizer to Nesterov. To determine the optimal outer learning rate, we follow “Scaling Laws for DiLoCo” sweeping over $\{0.2,0.4,0.6,0.8,1.0\}$, and we set the server momentum to $0.9$. Our experiment reveals:
>
> 1. **Memory Overhead:** LoRDO beats DiLoCo while maintaining a lower local memory overhead (DiLoCo requires full-rank AdamW). Furthermore, our standard LoRDO setup (averaging) *avoids storing* DiLoCo's additional server momentum term.
> 2. **Momenta Synchronization:** Disabling optimizer state communication harms both DiLoCo’s and LoRDO's performance (additionally see *opt_state_sync.pdf*).
> 3. **Outer Optimizer:** While Nesterov momentum provides improvement in the case of AdamW locally, this is not the case for low-rank optimizers where all results are strictly worse (see *lordo_outer_opt.pdf*), which requires further research. We have carried out preliminary investigations in this direction and show that the optimal hyperaparameters of DiLoCo’s Nesterov outer optimizer do not transfer to the low-rank setting, and the low-rank methods need to rotate the outer momenta (*lordo_outer_opt.pdf*).
>
> Prior work has also shown that communicating optimizer states improves training stability and machine learning performance [MT-DAO]. Nesterov being a special case of QHM implies that outer optimizer momentum could be implemented as a quasi-hyperbolic update plus normal momentum.
>
> ### **Q3: Addition of Full-Rank Signal Locally**
>
> You correctly identify that the workers combine the low-rank and full-rank signals in their local step. In the camera-ready version, we will make this point more explicit.
>
> ### **Q4: Second-Order Information for Projection**
>
> Besides the reasons already mentioned in Appendix E.3, we posit that second-order information would improve the signal for the projection matrix as the SVD on raw gradients ($g \approx -H p^{\star}$) gets dominated by the steep, high-curvature directions of $H$. Applying second-order information ($-H^{-1}g \approx p^{\star}$) mathematically cancels out this curvature distortion, forcing the SVD to project directly onto the subspace of true optimal steps. We leave a full investigation of this point as future work.
>
> ### **Presentation**
>
> In Figure 2, the top subplot is for 16M models. This will be corrected.
>
> ### **Closing**
>
> We hope these clarifications directly address your concerns. If you are satisfied with our responses, we kindly invite you to consider raising your score.

---

> > ### Author Rebuttal · Reviewer_4E3W · 2026-04-04
> >
> > Thank you for your response, my concerns have been addressed and I maintain my accept rating. That Nesterov harms performance in this setting is surprising, and warrants more investigation, however this is not neccessary for your paper.

---

> > > ### Author Response · Authors · 2026-04-06
> > >
> > > Dear Reviewer 4E3W,
> > >
> > > We are happy that you found our response to be satisfactory, and that you are maintaining your acceptance score. We will include the Nesterov experiments in the appendix of our submission, noting that future work on this point is needed.

---

### Decision · Program_Chairs · 2026-04-30

**Decision:**

Accept (regular)

**Comment:**

This paper introduces an approach that adapts low-rank optimizers for communication-efficient distributed training. The approach is original and improves communication overhead over competitive baselines.

The reviewer consensus leans positive, with strong appreciation for the clear failure-mode analysis of local low-rank projections. During the discussion and rebuttal phase, the authors successfully resolved the primary concerns raised by the committee, including: 1) memory and communication overhead, 2) DiLoCo Comparisons, and 3) Practical Validation.

Thus, overall, I support acceptance.